# Elevated TGF-β1 impairs synaptic and cognitive function through activation of Smad2/3-Sp1 pathway in AngII-related hypertension

Cuiping Guo[1,2,3,7], Wensheng Li[3,7], Yuanyuan Li[4,7], Yi Liu[3], Yacoubou Abdoul Razak Mahaman[3], Jianzhi Wang[1,2,3], Hongbin Luo[5], Rong Liu [ID][3,6], Hui Shen [ID][4✉] & Xiaochuan Wang [ID][1,2,3,6✉]

## Abstract

**Vascular dementia (VaD) is characterized by cognitive decline due to reduced cerebral blood flow, although its molecular mechanisms remain unclear. This study shows that angiotensin II (AngII) elevates blood pressure, reduces hippocampal blood flow, and impairs synaptic and cognitive function, which correlates with increased TGF-β1 levels. Overexpressing TGF-β1 in rats induces similar deficits, while its downregulation partially mitigates these effects, with the exception of hypoperfusion. Phosphorylation of Smad2/3, downstream of TGF-β1, is elevated in AngII-treated rats and TGF-β1-exposed neurons, and inhibiting Smad2/3 activation prevents synaptic damage. Additionally, phosphorylated Smad2/3 interacts more with the transcription factor Sp1 in hippocampal neurons of AngII-treated rats. Overexpression of Sp1 worsens synaptic and cognitive function, whereas Sp1 knockdown improves TGF-β1-induced impairments. These findings highlight TGF-β1 as a key mediator of AngII-induced cognitive deficits, beyond hypoperfusion, suggesting that targeting the TGF-β1/Smad2/3/Sp1 axis may offer therapeutic benefits for hypertension-related synaptic and cognitive dysfunction.**

**Keywords** AngII; TGF-β1; Sp1; Synaptic Dysfunction; Cognitive Impairments
**Subject Categories** Neuroscience; Signal Transduction

## Introduction

Cerebrovascular dementia can result from ischemic stroke, hemorrhagic stroke, and hypoxic-ischemic encephalopathy (Arvanitakis et al, 2019; Johnson, 2023; O'Brien and Thomas, 2015; Raz et al, 2016; Sommerlad et al, 2023). Recent studies indicate that vascular risk factors, especially hypertension, might influence the development of dementia (Kelly and Rothwell, 2020; Lespinasse et al, 2023; Takeda et al, 2020). The relationship between cerebrovascular dementia and the renin-angiotensin system (RAS) is widely acknowledged (Harrison et al, 2021; Royea et al, 2020). Angiotensin II (AngII) is a key component of the RAS which can cause excessive vasoconstriction, inflammation, altered cell dynamics, and increased extracellular matrix synthesis, together contributing to hypertension and vascular remodeling (Kumar et al, 2012; Marques et al, 2011; Takeda et al, 2020).

AngII can activate the transforming growth factor-beta (TGF-β) signaling pathway, a crucial regulator of cell survival, brain homeostasis, angiogenesis, memory, and neuronal plasticity (Border and Noble, 1998; Geara et al, 2009; Shabanian et al, 2022). In addition, increasing evidence suggests that TGF-β1 is implicated in detrimental processes related to aging and brain injury in the central nervous system, particularly in promoting astrocyte scarring (Caraci et al, 2011; Chen et al, 2023; Kerkering et al, 2023; Schneider et al, 2021). Fluctuations in TGF-β1 levels may contribute to cognitive disorders like Alzheimer's disease, schizophrenia, and depression (Caraci et al, 2011; He et al, 2021; Pan et al, 2022).

Considering these potential implications of TGF-β1, we aim to explore current research on the relationship of TGF-β1 with cognitive function. TGF-β1 binds to a heteromeric receptor complex, including TGF-β receptor type I (TGF-βRI) and type II (TGF-βRII). Upon activation, TGF-βRI phosphorylates the receptor-regulated Smad proteins, primarily Smad2 and Smad3 (Samy et al, 2016; Su et al, 2010). Phosphorylated Smads bind together to form a complex that enters the nucleus, binds to the Smad-binding elements, and recruits additional transcription regulators to control gene transcription (Miyazawa et al, 2024; Weiss and Attisano, 2013).

SP1 is a well-known transcription factor across various physiological and pathological processes (Convissar et al, 2019). Studies have indicated that Sp1 is upregulated in models of Huntington's disease (HD), and reducing its levels offers

[1]Institutes of Biomedical Sciences, School of Medicine, Hubei Key Laboratory of Cognitive and Affective Disorders, Jianghan University, 430056 Wuhan, China. [2]Co-innovation Center of Neuroregeneration, Nantong University, 226001 Nantong, China. [3]Department of Pathophysiology, School of Basic Medicine, Key Laboratory of Education Ministry/Hubei Province of China for Neurological Disorders, Tongji Medical College, Huazhong University of Science and Technology, 430030 Wuhan, China. [4]Laboratory of Neurobiology, School of Basic Medicine, Tianjin Medical University, Tianjin, China. [5]Health Science Center, HuBei Minzu University, 445000 Enshi, China. [6]Shenzhen Huazhong University of Science and Technology Research Institute, 518000 Shenzhen, China. [7]These authors contributed equally: Cuiping Guo, Wensheng Li, Yuanyuan Li. ✉E-mail: shenhui@tmu.edu.cn; wangxiaochuan@hust.edu.cn

neuroprotective benefits (Dunah et al, 2002; Niu et al, 2020). In addition, elevated Sp1 mRNA and protein levels in the cortex, hippocampus, and cerebellum of AD patients and transgenic mice exacerbate tau pathology, amyloid plaque formation, and cognitive decline (Citron et al, 2008; Citron et al, 2015; Villa et al, 2013). Therefore, it is crucial to investigate the role of Sp1 in cognitive deficits related to AngII type hypertension.

In this report, we demonstrate that TGF-β1 signaling contributes to AngII type hypertension-related cognitive impairments. Additionally, downregulating TGF-β1 ameliorated synaptic and cognitive impairments by inhibiting the Smad2/3 axis. AngII promoted phosphorylation and nuclear localization of Smad2/3 and significantly increased Smad2/3 binding to the transcriptional regulator Sp1. Furthermore, reducing Sp1 levels reversed the synaptic toxicity and memory loss induced by TGF-β1. Understanding the molecular mechanisms of TGF-β1 signaling is crucial for developing new therapies for neurological diseases associated with its dysregulation. Our findings offer new insights into the mechanisms of cognitive impairments in AngII-related hypertension and lay the groundwork for therapeutic strategies in vascular dementia management.

# Results

## Hypertension and cognitive impairments in AngII-treated rats

Hypertension is a recognized risk factor for cognitive decline (Duron and Hanon, 2008; Lis and Gaviria, 1997; Tzourio, 2007). Abnormal expression or activity of the renin-angiotensin system is closely associated with vascular remodeling and abnormal blood pressure (Royea et al, 2020). In our study, we administered rats with AngII for 28 days using a subcutaneous pump, which is capable of raising blood pressure and inducing hypertension. A schematic diagram of the experimental procedure is shown in Fig. 1A. Systolic blood pressure (SBP) was measured using tail-cuff plethysmography, and the results expectedly reveal a significant increase by $30 \pm 5$ mmHg in SBP in the AngII-treated rats compared to controls two weeks after use of the pump (Fig. 1B), with no significant differences in the body weight (Fig. 1C). Brain and hippocampal blood flow were assessed using 13N-NH3-PET rat brains dynamic imaging to measure the uptake of 13N-ammonia (Fig. EV1A). The results showed a decreased blood flow in the entire brain including both dorsal and ventral regions of the hippocampus in the AngII group (Fig. EV1B). These findings indicate that AngII administration significantly increased systolic blood pressure, leading to hypertension and a marked reduction in cerebral and hippocampal blood flow.

To better assess the hypertension-associated cognitive deficits, we performed a series of behavioral tests. The open field test assessing spontaneous activity revealed no significant differences between the AngII-treated and control groups in distance (Fig. 1D) and times across central areas (Fig. 1E). To assess anxiety, we used the elevated plus maze test (Fig. EV1C) and found that AngII-treated rats entered the open arms less frequently but entered the closed arms more often compared to controls (Fig. EV1D). In the Morris water maze test (Fig. 1F), which measures learning and memory, swim speed revealed no significant differences (Fig. 1F).

However, AngII-treated rats took a significantly longer time to locate the hidden platform compared to controls (Fig. 1G). On day 6, to assess spatial memory, the platform was removed, and AngII-treated rats spent significantly less time in the target quadrant (Fig. 1H) and crossed the platform area fewer times (Fig. 1I). We also performed fear conditioning test (Fig. 1J) to assess contextual learning and memory. The results showed significantly shorter freezing times in AngII-treated rats at both 2 (Fig. 1J) and 24 h (Fig. 1K). Collectively, these results suggest that AngII-related hypertension leads to cognitive deficits.

## AngII-related hypertension induced synaptic dysfunction

The hippocampus is a crucial component of the brain's limbic system that is involved in memory and spatial localization (Biegler et al, 2001; Dusek and Eichenbaum, 1997). Hippocampal synaptic plasticity, primarily characterized by long-term potentiation (LTP), is considered the cellular basis for learning and memory (Knierim, 2000). To investigate whether AngII-related hypertension mediates synaptic dysfunction we performed electrophysiological experiments. The results demonstrated that AngII-treated rats exhibit a decreased slope of field excitatory postsynaptic potentials (fEPSP) following high-frequency stimulation (HFS) compared to controls (Fig. 2A,B). To further investigate impairments in synaptic transmission along the CA3-CA1 Schaffer collateral pathway, we analyzed the paired-pulse ratio (PPR), a measure of transient presynaptic function plasticity. The PPR test results indicated no significant differences in presynaptic vesicle release between AngII-treated and control rats by inter-stimulus intervals of 50 ms (Fig. 2C,D). In contrast, whole-cell patch clamp recordings (Fig. 2E) of miniature excitatory postsynaptic potentials (mEPSCs) in the CA1 region showed that AngII treatment significantly reduced both the amplitude and frequency of mEPSCs (Fig. 2F,G). Recordings of miniature inhibitory postsynaptic potentials (mIPSCs) showed no significant differences (Fig. 2H,I). This suggests that AngII-related hypertension severely impairs excitatory synaptic transmission in the hippocampus. In addition, we examined the dendritic architecture of hippocampal CA1 neurons (Fig. 2J). The results from Golgi staining showed that AngII resulted in an obvious decrease in the dendritic complexity compared to the control group (Fig. 2K), as well as a significant decrease in the total dendritic length (Fig. 2L) and the dendritic spine density (Fig. 2M). These results implied that AngII-related hypertension induced synaptic disorders.

## AngII-related hypertension significantly upregulated the TGF-β1 signaling

To explore the role of AngII-related hypertension in cognitive deficits, we performed bioinformatics analyses aiming to identify marker genes associated with AngII models. In the GSE47529 dataset, which was accessed from the Gene Expression Omnibus (GEO) database. After normalizing the matrices and addressing batch differences, the gene expression density distributions were consistent across datasets, suggesting reliable data sources for further analysis (Fig. EV2A). In the AngII-related dataset GSE47529, the bioinformatics analysis results showed significant expression levels of differentially expressed genes (DEGs) (Fig. EV2B,C). Functional enrichment analysis was conducted

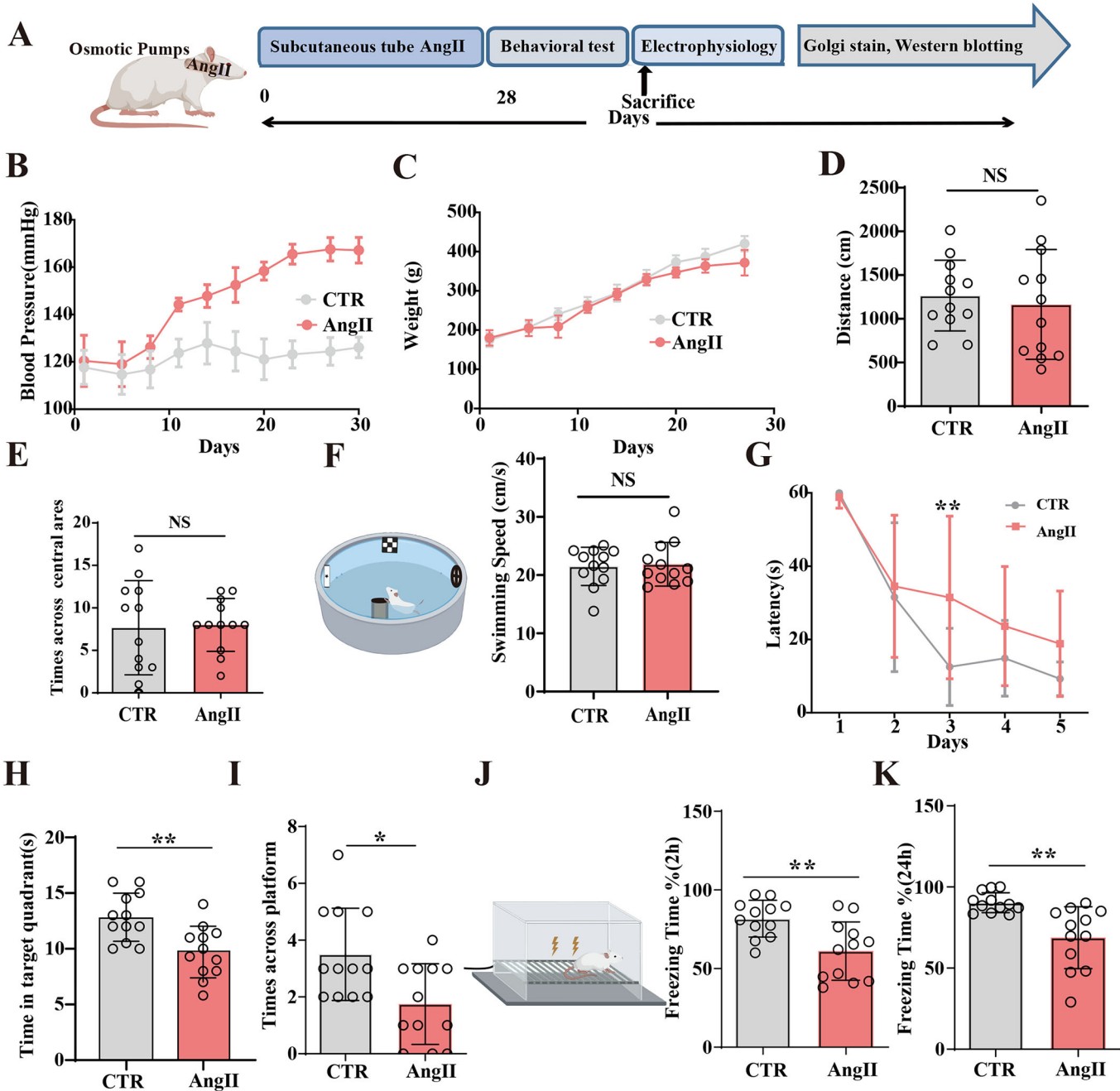

**Figure 1. Hypertension and cognitive impairments in AngII-treated rats.**

(A) Schematic representation of the outline of the study: Subcutaneous tube administration of AngII induced hypertension in rats. Blood pressure, cerebral blood flow, behavior, electrophysiology, molecular biology, and morphology were measured. (B) Tail-cuff plethysmography was used to measure SBP ($n = 12$). (C) All rats were weighed during the 4 weeks period of the study, ($n = 12$). The open field test assesses the total distance covered ($n = 12$) (D) and entries into central areas ($n = 12$), (E) in the two groups. In the Morris water maze test, the swimming speed ($n = 12$), (F), the latency to find the hidden platform (G) from day 1 to day 5 was measured, and the spatial memory was tested on the 6th day by removing the platform ($n = 12$), ($P = 0.0071$), and the time spent in the target quadrant ($n = 12$), ($P = 0.0025$), (H) and the number of crossing the position of the target platform ($n = 12$), ($P = 0.0102$), (I) were measured. Fear conditioning test was used to measure the contextual memory: freezing duration was measured during the 3-min memory test at 2 h ($P = 0.0037$), (J) and 24 h ($P = 0.0012$), (K) after conditioning ($n = 12$). Data are presented as mean ± SD. A two-tailed Student's *T* test was used for statistical analysis in (D–F, H–K). Two-way ANOVA of Sidak's multiple comparisons test was used for statistical analysis in (G). *$P < 0.05$, **$P < 0.01$, versus Control group. Source data are available online for this figure.

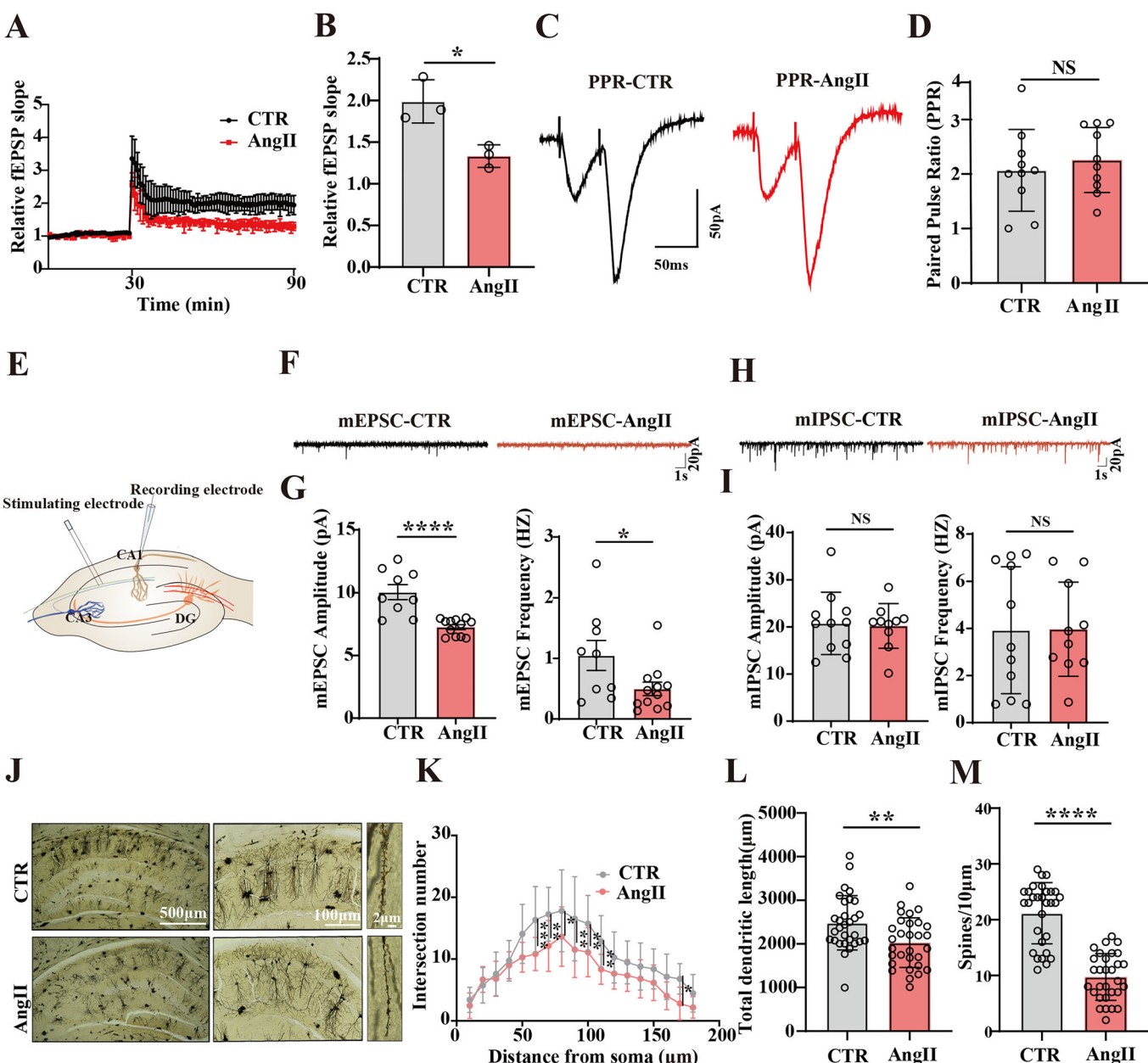

**Figure 2. AngII-related hypertension induced synaptic dysfunction.**

Hippocampal CA3-CA1 LTP (A) and its quantification (P = 0.0176) (B) were recorded by using the MED64 system (n = 3). (C) The representative image showed a paired-pulse ratio (PPR) test in hippocampal slices. For pairwise pulse ratio experiments, paired stimuli by inter-stimulus intervals of 50 ms were delivered (C) and the ratio was calculated as the amplitude ratio (EPSC2/EPSC1) (D), (n = 10 recordings/4 rats per group). (E) Rat hippocampal CA1 neurons were subjected to whole-cell patch-clamp recording. (F, G) Amplitude (P < 0.0001) and frequency (P = 0.0393) of mEPSCs were measured (n = 9–12 recordings/4 rats per group). (H) Rat hippocampal CA1 neurons were subjected to whole-cell patch-clamp recording of mIPSCs and amplitude and frequency (I) of mIPSCs were measured (n = 10–11 recordings/4 rats per group). (J) Representative dendrites from Golgi-impregnated hippocampus neurons, Scale bar = 500 μm. Sholl analysis (Row6 P = 0.0004; Row7 P = 0.0012; Row8 P = 0.0223; Row9 P = 0.0019; Row10 P = 0.0061; Row11 P = 0.0082; Row17 P = 0.0353), (K) (n = 15), quantitative analyses of dendritic length (P = 0.0047) (scale bar = 100 μm, n = 30), (L) and averaged spine density (P < 0.0001), (mean spine number per 10-μm dendrite segment), (scale bar = 2 μm, n = 30) (M). Data are presented as Mean ± SD. A two-tailed Student's T-test was used for statistical analysis in (B, D, G, I, L, M). Two-way ANOVA of Sidak's multiple comparisons test was used for statistical analysis in (K). *P < 0.05, **P < 0.01, ***P < 0.001, ****P < 0.0001, versus Control group. Source data are available online for this figure.

using both GO (Fig. EV2D) and KEGG (Fig. EV2E,F) pathway enrichment analysis. 27 genes were identified upon intersecting the DEGs with core enrichment genes in the "Neuroactive ligand-receptor interaction" and "Cellular senescence"(Fig. 3A). Moreover,

functional enrichment GO pathway analysis was conducted with the intersected genes (Fig. 3B). Spearman rank correlation analysis was used to identify the relationships among the 27 genes (Fig. 3C). Consistently, the hub gene, TGF-β1 was identified through the

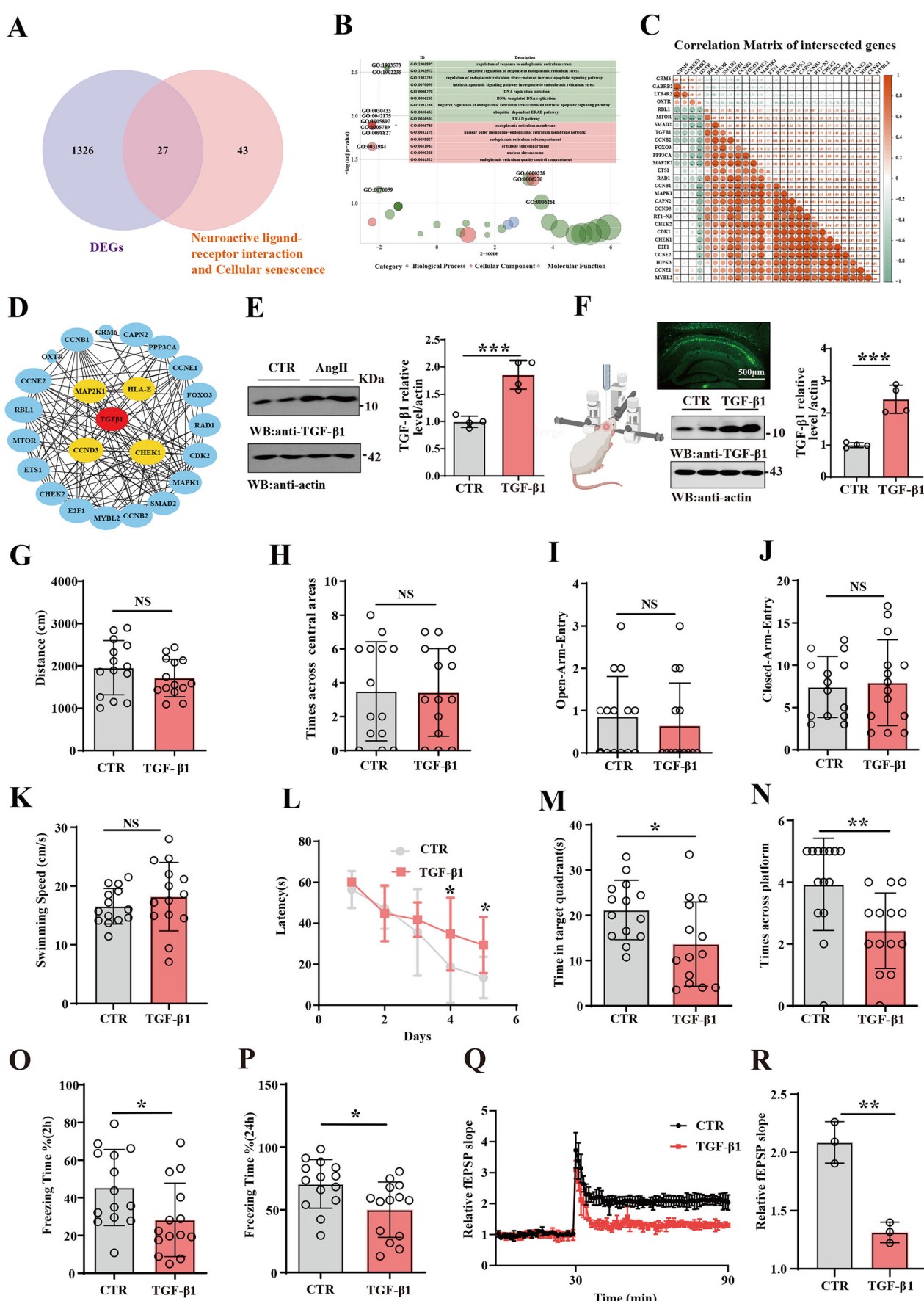

**Figure 3. Overexpression of TGF-β1 resulted in synaptic damage and learning and memory impairments.**

(A) Venn diagram illustrates the intersection between DEGs (|Log2FoldChange| > 0.35 and P < 0.05) and core enrichment genes in the "Neuroactive ligand-receptor interaction" and "Cellular senescence" pathways. (B) GO enrichment analysis results for the intersected genes from the Venn diagram. (C) Spearman rank correlation analysis of the intersected genes. (D) Protein–Protein Interaction (PPI) network of intersected genes, highlighting MAP2KA, HLA-E, CCND3, CHEK1 and TGF-β1 as hub genes. (E) Brain tissues (hippocampus CA1 region) were homogenized, and TGF-β1 protein level was detected by immunoblotting. Actin was used as a loading control. Quantitative analysis of the TGF-β1 level (P = 0.0009) (n = 4). (F) TGF-β1 virus was injected into the hippocampal CA1 region of the SD rats, and GFP expression (green) was observed 4 weeks after injection. Hippocampus CA1 region was homogenized, and TGF-β1 protein level was detected by immunoblotting. Actin was used as a loading control; Quantitative analysis of the TGF-β1 level (P = 0.0007) (n = 4). The open field test was measured to the total distance covered (G) and entries into central areas (H) in the two groups (n = 14). The high plus maze test was performed to measure the entries to the open (I) and closed arms (J), (n = 14). (K–N) Morris Water Maze Test: swimming speed (K), latency to find the hidden platform, (Row4, P = 0.0105; Row5, P = 0.0109), (L) from day 1 to day 5 was recorded. Spatial memory was assessed on the 6th day by measuring the time spent in the target quadrant (P = 0.0198), (n = 14), (M) and the number of target platform crossings (P = 0.0073), (n = 14) (N). (O, P) Fear conditioning test was used to assess the contextual memory, and the freezing duration was measured during the 3-min memory test at 2 h (P = 0.0308), (O) and 24 h (P = 0.0148), (P) after conditioning (n = 14). (Q) Hippocampal CA3-CA1 LTP and its quantification (P = 0.0025), (n = 3), (R) were recorded by using the MED64 system. Data are presented as mean ± SD. A two-tailed Student's T test was used for statistical analysis in (E–K, M–P, R). Two-way ANOVA of Sidak's multiple comparisons test was used for statistical analysis in (L). *P < 0.05, **P < 0.01, ***P < 0.001versus Control group. Source data are available online for this figure.

Protein–Protein Interaction (PPI) network analysis with connectivity degree among 27 intersected genes (Fig. 3D). At the same time, the Elisa assay results showed that the AngII levels were increased both in hippocampus (Fig. EV3A) and in plasma (Fig. EV3B). Consistent with this, The TGF-β1 levels were increased in plasma (Fig. EV3C) and western blotting analysis revealed significantly higher levels of TGF-β1 in hippocampus of the AngII group compared to controls (Fig. 3E). Besides, we have performed the primary hippocampal neuronal culture with or without AngII 0.1 μM (Fig. EV3D) for 6 h, oxygen-glucose deprivation (OGD) for 12 h (Fig. EV3F) and CoCl₂ (Fig. EV3H), a drug for chemical hypoxia model(Pecoraro, Pinto et al, 2018; Zhang, Ma et al, 2015), 100 μmol/L for 6 h. We found that AngII does not affect the levels of TGF-β1 in the primary hippocampal neurons (Fig. EV3E), while OGD and CoCl₂ both increased the levels of TGF-β1 (Fig. EV3G, I). Collectively, these findings suggest that AngII may contribute to hypertension-related cognitive impairments through the activation of the TGF-β1 signaling pathway.

## Overexpression of TGF-β1 resulted in synaptic damage and learning and memory impairments

To further evaluate the impact of TGF-β1 on cognitive function, we conducted bilateral intra-hippocampal CA1 injections with AAV-TGF-β1 and vector virus serving as the control. GFP expression in the hippocampal CA1 was confirmed by fluorescence microscopy after 4 weeks and western blotting analysis revealed a significant increase in TGF-β1 levels in the AAV-TGF-β1 group compared to controls (Fig. 3F). Then, we carried out the behavioral tests. The open-field test revealed no significant differences in distance (Fig. 3G) and times across central areas (Fig. 3H), as well as no significant differences in elevated plus maze (Fig. 3I,J). The Morris water maze data indicated that swim speed revealed no significant differences (Fig. 3K). However, TGF-β1-treated rats had significantly increased latency in finding the hidden platform compared to controls (Fig. 3L). On day 6, the probe test showed TGF-β1-treated rats spent significantly less time in the target quadrant (Fig. 3M) and crossed the platform fewer times than the controls (Fig. 3N). The fear conditioning test revealed significantly reduced freezing durations in the TGF-β1 group at both 2 (Fig. 3O) and 24 h (Fig. 3P). These findings collectively indicate that TGF-β1 may contribute to learning and memory impairments. In line with this,

results from electrophysiology experiments showed that TGF-β1 reduces the fEPSP after high-frequency stimulation compared to the control group (Fig. 3Q,R). These results further support that TGF-β1 seriously causes synaptic damage and learning and memory impairment.

## Downregulation of TGF-β1 improved AngII-induced synaptic and cognitive deficits

To investigate the significance of TGF-β1 implication in AngII-related cognitive impairments, SD rats were randomly assigned to three groups (Control, AngII and AngII+Sh-TGF-β1groups), a schematic diagram of the experimental process is provided in Fig. 4A. Systolic blood pressure was measured using tail-cuff plethysmography, and the data revealed a significant increase of 30 ± 5 mmHg in the AngII group compared to controls two weeks after use of the pump. However, there were no significant differences between the AngII+Sh-TGF-β1 and AngII groups (Fig. 4B). Dynamic imaging of the rat brains using 13N-NH3-PET demonstrated a decrease in blood flow in both the dorsal and ventral hippocampal regions of the AngII group. However, no significant differences were observed between the AngII+Sh-TGF-β1 and AngII groups (Fig. EV4A,B). These suggest that TGF-β1 may be another key molecule in AngII-induced cognitive dysfunction besides hippocampal hypoperfusion.

Subsequently, Morris Water Maze test showed that AngII+Sh-TGF-β1 rats had significantly shorter latency in locating the hidden platform when compared with the AngII group (Fig. 4C). On day 6, spatial memory was assessed by removing the platform, and the results revealed that AngII+Sh-TGF-β1 rats crossed the position of the hidden platform more frequently than the AngII group (Fig. 4E), but no significant differences in spending time in the target quadrant (Fig. 4D). Additionally, the fear conditioning test indicated that the freezing duration within 2 and 24 h (Fig. 4F,G) were significantly longer in the AngII+Sh-TGF-β1 group compared to the AngII group.

In addition, electrophysiology experiments demonstrated that downregulating TGF-β1 improved the slope of the field excitatory postsynaptic potentials following high-frequency stimulation (Fig. 4H,I). The results of whole-cell patch clamp recordings measuring the mEPSCs of glutamate receptors in the CA1 region (Fig. 4J) indicate that knocking down TGF-β1 rescued excitatory synaptic transmission in the hippocampus (Fig. 4K,L). Moreover,

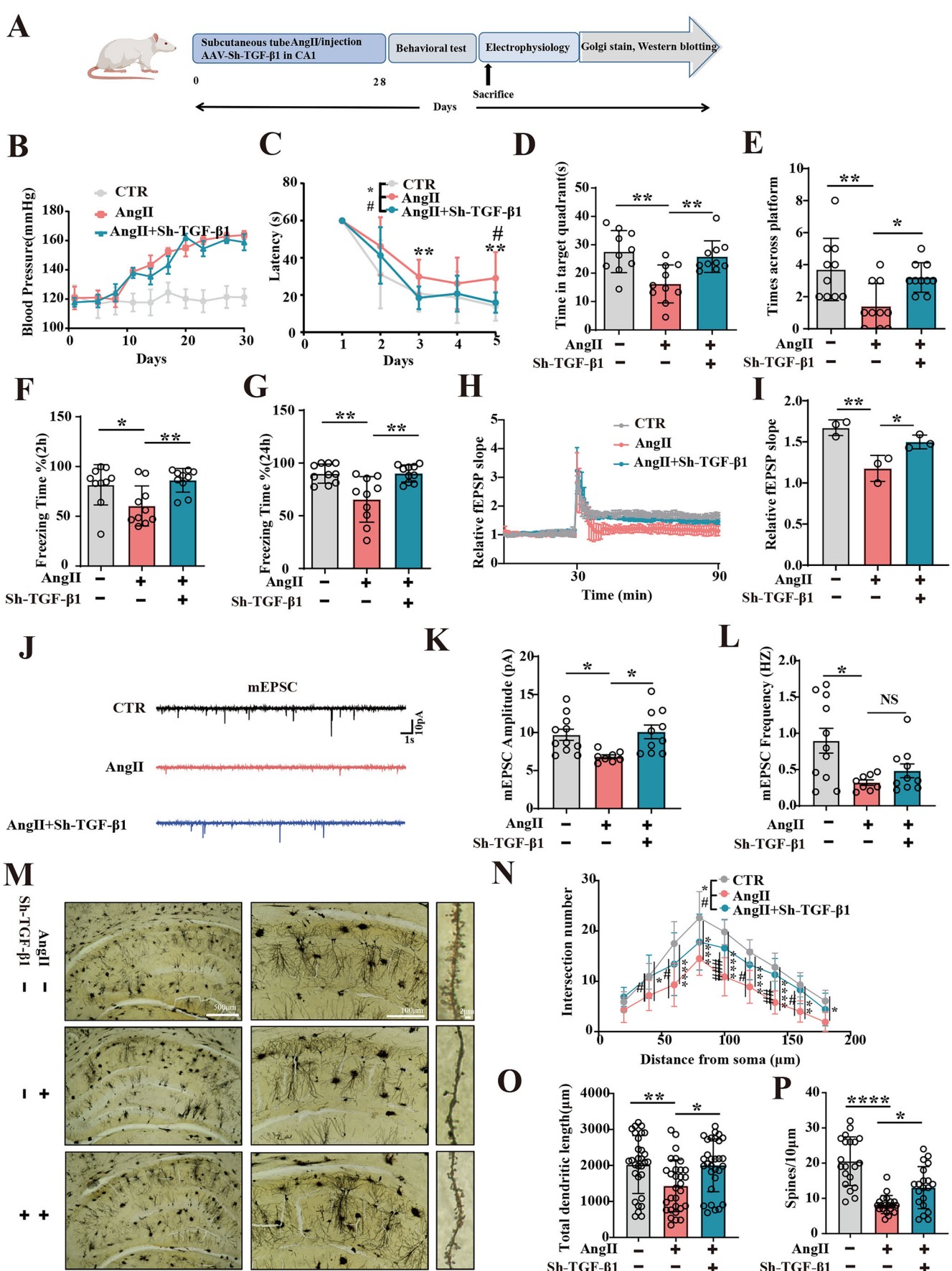

**Figure 4.  Downregulation of TGF-β1 improved AngII-induced synaptic and cognitive deficits.**

(A) Schematic representation of the outline of the study: Subcutaneous tube administration of AngII and bilateral intra-hippocampal CA1 injections of AAV-Sh-TGF-β1 virus. Blood pressure, cerebral blood flow, behavior, electrophysiology, molecular biology, and morphology were evaluated. (B) The tail-cuff plethysmography was used to measure SBP ($n = 10$). (C–E) Morris Water Maze Test: latency to find the hidden platform (Row3, $P = 0.0066$; Row5, $P = 0.0072$, $P = 0.0218$), (C) from day 1 to day 5 was recorded, and spatial memory was assessed by measuring the time spent in the target quadrant ($P = 0.0018$, $P = 0.0082$), (D) and the number of target platform crossings ($P = 0.0051$; $P = 0.0309$), (E) on the 6th day ($n = 10$). (F, G) Fear conditioning test was used to measure the contextual memory, where the freezing duration was measured during the 3-min memory test at 2 h ($P = 0.0330$; $P = 0.0089$), (F) and 24 h ($P = 0.0020$; $P = 0.0019$), (G) after conditioning, ($n = 10$). (H) Hippocampal CA3-CA1 LTP and its quantification (I) were recorded by using the MED64 system ($P = 0.0049$, $P = 0.0351$), ($n = 3$). (J) Rat hippocampal CA1 neurons were subjected to whole-cell patch-clamp recording of mEPSCs, the amplitude ($P = 0.0295$, $P = 0.0148$), (K) and frequency ($P = 0.0114$), (L) of mEPSCs were measured ($n = 8$–$11$ recordings/4 rats per group). (M) Representative dendrites from Golgi-impregnated hippocampus neurons, scale bar = 500 μm. Sholl analysis, ($n = 10$), (Row4 $P = 0.0419$, $P = 0.0492$; Row6 $P < 0.0001$, $P = 0.0252$; Row8 $P < 0.0001$, $P = 0.0901$; Row10 $P < 0.0001$, $P = 0.0010$; Row12 $P < 0.0001$, $P = 0.0147$; Row14 $P < 0.0001$, $P = 0.0015$; Row16 $P = 0.0024$, $P = 0.0177$; Row18 $P = 0.0211$), (N) quantitative analyses of dendritic length (scale bar = 100 μm, $n = 30$), ($P = 0.0089$, $P = 0.0124$), (O) and averaged spine density (mean spine number per 10-μm dendrite segment), (scale bar = 2 μm, $n = 20$), ($P < 0.0001$, $P = 0.0203$), (P). Data are presented as mean ± SD. One-way ANOVA with Tukey's multiple comparisons test was used for statistical analysis in (D–G, I, K, L, O, P). Two-way ANOVA with Tukey's multiple comparisons test for (C, N). *$P < 0.05$, #$P < 0.05$, **$P < 0.01$, ##$P < 0.01$, ###$P < 0.001$, ****$P < 0.0001$, versus AngII group. Source data are available online for this figure.

we examined the dendritic architecture of hippocampal CA1 neurons (Fig. 4M). The results from Golgi staining showed that AngII+Sh-TGF-β1 rats resulted in an obvious increase in the dendritic complexity compared to the AngII group (Fig. 4N), as well as a significant increase in the total dendritic length (Fig. 4O) and the dendritic spine density (Fig. 4P). Together, these findings strongly support that TGF-β1 plays a significant role in the synaptic dysfunction and cognitive impairments associated with AngII-related hypertension.

## TGF-β1 activates the Smad2/3 pathway, leading to neuronal damage

Our findings suggest that TGF-β1 upregulation may at least in part mediate AngII-related synaptic dysfunction. We found that the phosphorylation of Smad2/3 complex, which is downstream of the TGF-β1 pathway, was significantly increased following TGF-β1 treated- hippocampal primary neurons (Fig. 5A, B). Coherently, the phosphorylation of Smad2/3 was significantly increased in AngII rats (Fig. 5C,D). Western blotting analysis showed that treatment of hippocampal neurons with 0.5 μM SIS₃ HCl, a novel specific inhibitor of Smad3 that inhibits Smad3 phosphorylation, decreased the level of phosphorylated Smad2/3, while total Smad2/3 levels remained unchanged (Fig. 5E,F). Similarly, the dendritic morphology of primary hippocampal neurons was examined using GFP (Fig. 5G). The results indicated that inhibition of Smad3 phosphorylation significantly restored the complexity and length of dendrites in TGF-β1-treated neurons (Fig. 5H,I). The phosphorylated Smad2/3 complex translocates to the nucleus and regulates transcription of target genes (Miyazawa et al, 2024). Moreover, we isolated nuclear components and western blotting revealed an increased presence of phosphorylated Smad2/3 in the nucleus of hippocampal lysate in AngII rats (Fig. 5J,K). Consequently, we detected the downstream factors such as the specificity protein 1 (Sp1), IRF-7, c-Jun, c-Fos. After extracting the nuclear components, co-immunoprecipitation was used to detect the interaction (Fig. 5L). The findings exhibit a significantly increased interaction between these Smad2/3-Sp1 proteins in the AngII group (Fig. 5M). We have further investigated the effect of Smad2/3 on the transcriptional regulation of Sp1. Two plasmids with point mutations at the Smad3 Ser423/425 site, S423/425D (phosphorylation-activated state) and S423/425A (phosphorylation-inhibited state), were transfected into the N2A cell line. Then, we carried out

Chip-qPCR and detected the relationship between Sp1 and its relevant target genes *BACE1* and *HSP70* (Bevilacqua et al, 1997; Christensen et al, 2004; Morgan, 1989; Nakano et al, 2024; Nong et al, 2022), and found that mRNA levels of *HSP70* and *BACE1* significantly increased in the S423/425D group, while mRNA levels of *HSP70* were significantly decreased in the S423/425A group (Fig. 5N). Therefore, these data suggest that Sp1 may be an important factor potentially regulating neuronal impairment.

## Transcription factor Sp1 mediated synaptic dysfunction and cognitive deficits

To explore the role of Sp1 in cognitive deficits, we performed bioinformatics analyses aiming to identify Sp1-associated genes associated with neurological-related diseases. Datasets GSE126500, GSE165771, GSE37935, and GSE31628 were accessed from the Gene Expression Omnibus database (Fig. EV5A,B). A total of 65 co-DEGs were identified (Fig. 6A,B). KEGG pathway enrichment analysis of these DEGs indicated enrichment in the Sp1 signaling pathway, including a variety of neurodegenerative disorders, such as Parkinson disease, Alzheimer disease, Huntington disease, Spinocerebellar ataxia, etc (Fig. 6C), which suggest that alterations in the transcription factor Sp1 may play a critical role in neurodegenerative disorders. To explore the biological role of Sp1 in neurons, primary hippocampal neurons transduced with LV-Sp1 virus were analyzed for dendritic morphology using mCherry (Fig. 6D). Interestingly, Sp1 overexpression led to a significant decrease in the dendritic complexity (Fig. 6E), as well as a decrease in the total dendritic length (Fig. 6F) and the dendritic spines density (Fig. 6G). We subsequently conducted to assess cognitive functions. We conducted bilateral intra-hippocampal CA1 injections with LV-Sp1 virus and vector virus serving as the control. In the Morris Water Maze test, it showed no significant differences in swim speed (Fig. 6H), but Sp1 rats showed a significantly increased latency to find the hidden platform during the learning phase (Fig. 6I). During the probe test, a remarkably decreased time in the target quadrant (Fig. 6J) and target platform crossings (Fig. 6K) were observed in the Sp1 rats. The Fear condition revealed a significantly reduced freezing duration in the Sp1 group within 2 h (Fig. 6L) and 24 h (Fig. 6M). Moreover, electrophysiological experiments demonstrated that when compared to control, Sp1 rats exhibited a decreased slope of the fEPSP following HFS (Fig. 6N,O). These findings indicate that Sp1 severely impairs synaptic function in the hippocampus, suggesting that Sp1 may partly mediate the TGF-β1-induced synaptic dysfunction and cognitive deficits.

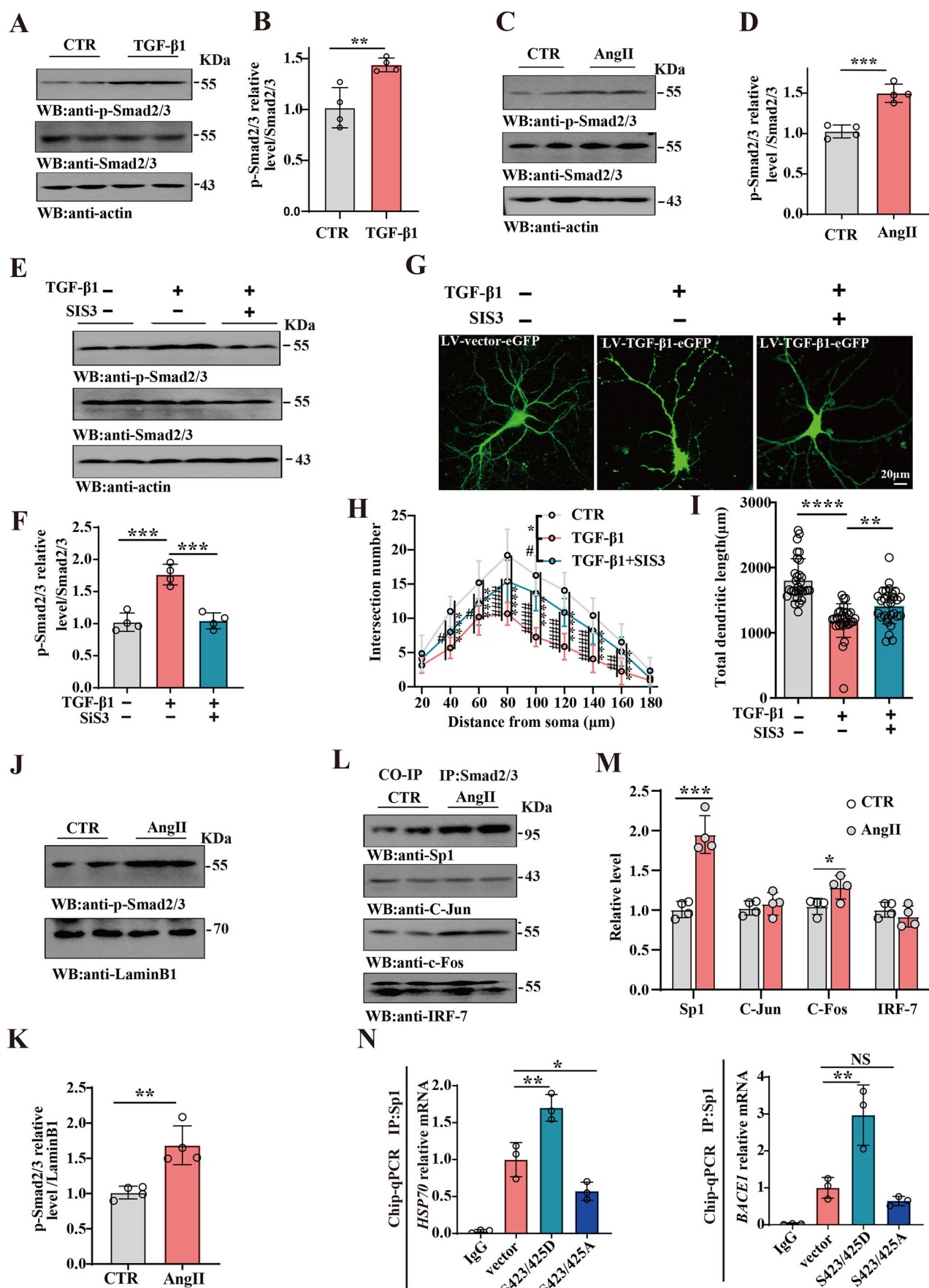

Figure 5.  TGF-β1 upregulated the Smad2/3 pathway leading to neuronal damage.

(A, C) Brain tissues (hippocampus region) were homogenized, and p-Smad2/3 and Smad2/3 protein levels were detected by immunoblotting. Actin was used as a loading control. (B, D) Quantitative analysis of the p-Smad2/3/Smad2/3 in TGF-β1 group ($P = 0.0068$), (B) and in AngII group ($P = 0.0005$), ($n = 4$), (D) (E) Hippocampal primary neurons were treated with compound SIS$_3$ HCl and LV-TGF-β1. Western blotting showed the p-Smad2/3 and Smad2/3 levels after treatment of hippocampal neurons with SIS$_3$ 0.5 μmol. Actin was used as a loading control. (F) Quantitative analysis of the p-Smad2/3/Smad2/3 ($P = 0.0001$, $P = 0.0001$), ($n = 4$). (G) The GFP was measured by immunofluorescence. (H) Sholl analysis (scale bar $= 20$ μm $n = 10$), (Row4 $P < 0.0001$, $P = 0.0168$; Row6 $P < 0.0001$, $P = 0.0487$; Row8 $P < 0.0001$ $P < 0.0001$; Row10 $P < 0.0001$, $P < 0.0001$; Row12 $P < 0.0001$, $P < 0.0001$; Row14 $P < 0.0001$, $P < 0.0001$; Row16 $P < 0.0001$, $P = 0.0022$), (I) quantitative analyses of dendritic length ($P < 0.0001$, $P = 0.0095$) ($n = 30$). (J) Nuclear protein components from hippocampal tissue were extracted, and p-Smad2/3 was detected by immunoblotting, laminB1 was used as a loading control. Quantitative analysis of the p-Smad2/3 ($P = 0.0036$) ($n = 4$). (K) Nuclear protein components from hippocampal tissue were extracted, (L, M) Co-IP test was used to examine protein Smad2/3-Sp1/c-Jun/c-Fos/IRF-7 interaction, ($P = 0.0003$, $P = 0.0365$), ($n = 4$). (N) Two plasmids with point mutations at the Smad3 Ser423/425 site, S423/425D (phosphorylation-activated state) and S423/425A (phosphorylation-inhibited state), were transfected into the N2A cell line. HSP70 ($P = 0.0029$, $P = 0.0440$) and BACE1 ($P = 0.0025$) genes were validated through Chip assay, ($n = 3$). Data are presented as mean ± SD. A two-tailed Student's T test was used for statistical analysis in (B, D, K, M). One-way ANOVA with Tukey's multiple comparisons test was used for statistical analysis in (F, I, N). Two-way ANOVA with Tukey's multiple comparisons test was used for statistical analysis in (H). *$P < 0.05$, #$P < 0.05$, ##$P < 0.01$, **$P < 0.01$, ***$P < 0.001$, ####$P < 0.001$, ****$P < 0.001$, versus Control or TGF-β1 group. Source data are available online for this figure.

## Downregulation of Sp1 improved TGF-β1-induced synaptic and cognitive deficits

To determine whether Sp1 plays a crucial role in TGF-β1-induced synaptic impairments or not, primary hippocampal neurons were co-transduced with LV-TGF-β1 and LV-Sh-Sp1 viruses. GFP and mCherry were then used to examine dendritic morphology (Fig. 7A). Interestingly, Sh-Sp1 led to a significant increase in the dendritic complexity (Fig. 7B), as well as an increase in the total dendritic length (Fig. 7C). To further evaluate the role of Sp1, we conducted bilateral intra-hippocampal CA1 injections to rats with LV-TGF-β1 virus and LV-Sh-Sp1 virus. SD rats were randomly assigned to three groups (Control, TGF-β1 and TGF-β1 + Sh-Sp1 groups). Fluorescence microscopy confirmed the expression of mCherry and GFP in the hippocampal CA1 region (Fig. 7D). Subsequently, Morris Water Maze test showed that TGF-β1 + Sh-Sp1 rats had a significantly shorter latency to localize the hidden platform compared to the TGF-β1 group (Fig. 7E). Moreover, the TGF-β1 + Sh-Sp1 group spent significantly more time in the target quadrant (Fig. 7F) and crossed the former platform location more frequently (Fig. 7G) than the TGF-β1 group. The Fear condition test revealed that the freezing duration within 2 h was significantly longer in the TGF-β1 + Sh-Sp1 group compared to the TGF-β1 group (Fig. 7H), with no significant differences noted at 24 h (Fig. 7I). Additionally, electrophysiological experiments indicated that downregulation of Sp1 enhanced the slope of fEPSP following high-frequency stimulation, compared to the TGF-β1 group (Fig. 7J,K). Consistently, whole-cell patch-clamp recording of mEPSCs in primary hippocampal CA1 neurons (Fig. 7L) showed that downregulation of Sp1 significantly increased the amplitude (Fig. 7M), but no significant changes in the frequency of mEPSCs (Fig. 7N). Collectively, these findings strongly support the hypothesis that TGF-β1-upregulated Sp1 mediates synaptic and cognitive impairments.

## Discussion

Hypertension is the most prevalent cause of vascular cognitive impairment. Excessive salt intake and the renin-angiotensin system (RAS) activation are both major causes of hypertension. Previous study has shown that a high-salt diet leads to endothelial NO

deficiency, resulting in calpain denitrosylation, which in turn activates Cdk5 and tau phosphorylation in neurons, finally leading to cognitive dysfunction (Faraco, Hochrainer et al, 2019). However, the pathogenesis of AngII type hypertension-related dementia is largely unclear. We have here elucidated the abnormal upregulation of TGF-β1 in AngII-related hypertension, which correlates with neuronal damage and synaptic dysfunction in the rat hippocampus. Interestingly, downregulation of TGF-β1 significantly improved synaptic damage and cognitive impairments. Increased TGF-β1 leads to elevated Smad2/3 phosphorylation resulting in the Smad2/3 then translocation into the nucleus, enhancing the activity of the transcription factor Sp1, which mediates synaptic and cognitive deficits. Thus, our study not only uncovers a novel role of TGF-β1 signaling in the pathophysiology and behavioral abnormalities of AngII-related hypertension but also highlights the potential of targeting TGF-β1 as a therapeutic strategy for treating hypertension-related cognitive deficits besides hippocampal hypoperfusion.

The RAS plays a key role in the development of hypertension and end-organ damage (Arnold et al, 2013; Navar et al, 2011; Tran et al, 2022). The effector molecule of the RAS, AngII is a potent vasoconstrictor and a crucial factor in the development of hypertension, which also acts as a potent pro-inflammatory mediator, stimulating the production of inflammatory cytokines and inducing oxidative stress via the AT receptors. This affects the vascular system, kidneys, and brain, ultimately promoting hypertension (Lelis et al, 2019; Saravi et al, 2021; Tran et al, 2022). In our study, a 28-day administration of AngII via a subcutaneous tube significantly increased blood pressure and induced hypertension. These data demonstrate that AngII-related hypertension impairs synaptic transmission and cognitive function possibly by over-activating TGF-β1 signaling.

TGF-β1 activates the Smads complex which promotes the phosphorylation and subsequent activation of the downstream Smad pathway. The activated Smads translocate into the nucleus, where they bind to transcription factor promoters and cofactors, initiating DNA transcription (Trinh et al, 2022). In the current study, TGF-β1 levels were found to be significantly increased in AngII-treated rats, thus serving as a marker for synaptic dysfunction and cognitive deficits. Downregulation of TGF-β1 alleviated these AngII-induced impairments in addition to hippocampal hypoperfusion, suggesting that TGF-β1 upregulation

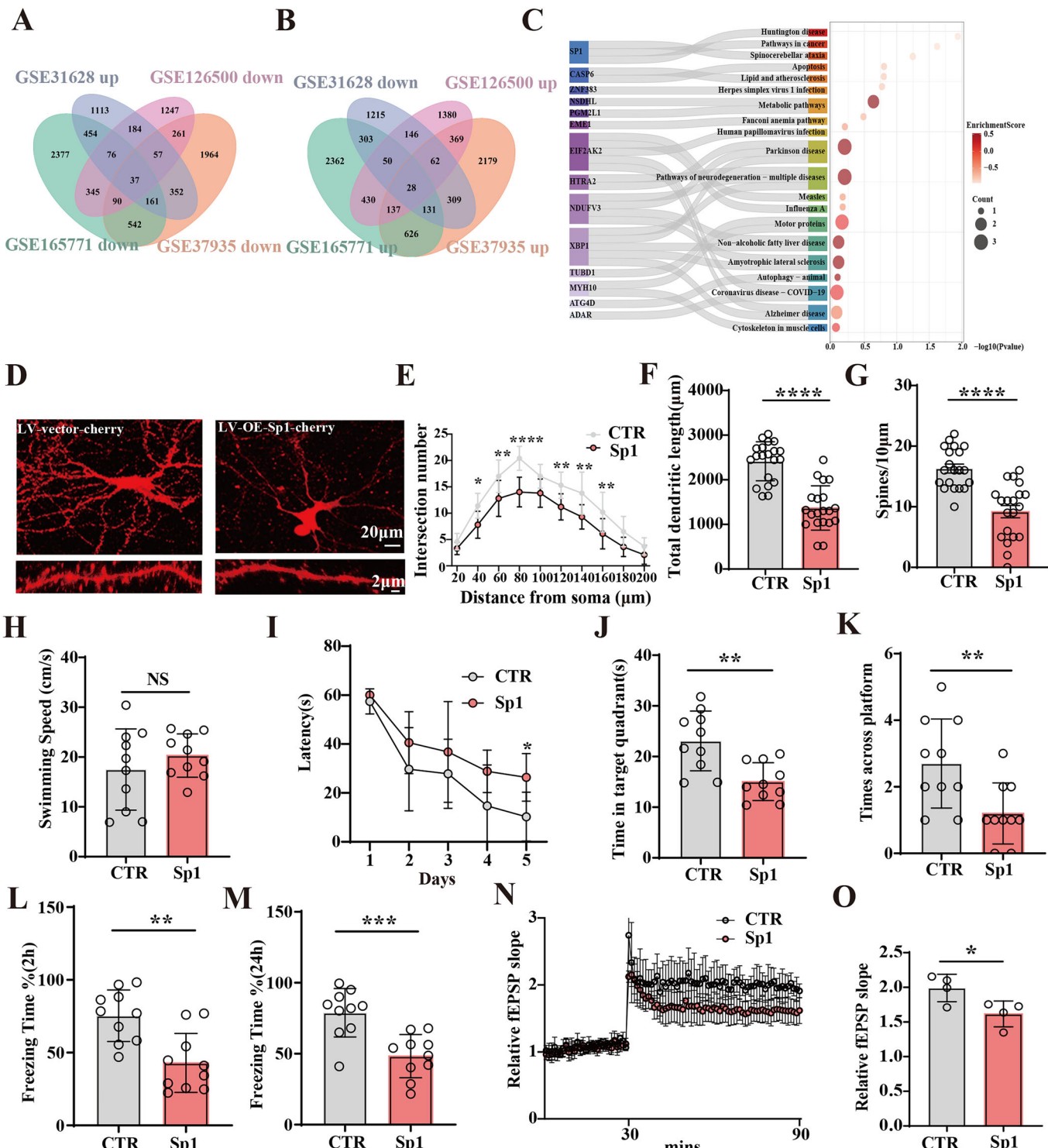

contributes to the pathogenesis of AngII-related hypertension. This upregulation promotes the entry of the phosphorylated Smad2/3 complex into the nucleus, where they bind to Sp1 instead of the other transcription factors IRF-7, c-Jun, c-Fos.

It has been previously reported that Sp1 dysfunction could significantly impact neurodegenerative disorders (Citron et al, 2008, 2015; Dunah et al, 2002; Niu et al, 2020). Consistent with this,

our findings demonstrate that Sp1 knockdown reversed the cognitive impairments induced by TGF-β1. Conversely, Sp1 overexpression in wild-type rats led to severe synaptic dysfunction in the hippocampal CA1 region, and consequently, cognitive deficits. Furthermore, downregulation of Sp1 significantly mitigated the detrimental effects of TGF-β1 on synaptic function, strongly supporting the notion that an increase in Sp1 is essential for the

**Figure 6. Sp1 mediated the synaptic dysfunction and cognitive deficits.**

(A) Venn diagram showing the intersection of downregulated DEGs in GSE126500, GSE165771, GSE37935 (knockdown of Sp1) and upregulated DEGs in GSE31628 (overexpression of Sp1). (B) Venn diagram showing the intersection of upregulated DEGs in GSE126500, GSE165771, GSE37935 (knockdown of Sp1) and downregulated DEGs in GSE31628 (overexpression of Sp1). (C) KEGG enrichment analysis results for DEGs. (D) Hippocampal primary neurons were administered with LV-vector or LV-Sp1 virus. (E) Sholl analysis (scale bar = 20 μm, $n = 10$), (Row4 $P = 0.0237$; Row6 $P = 0.0042$; Row8 $P < 0.0001$; Row12 $P = 0.0057$; Row14 $P = 0.0016$; Row16 $P = 0.0057$). (F) Quantitative analyses of dendritic length, ($P < 0.0001$), ($n = 20$). (G) Averaged spine density (mean spine number per 10-μm dendrite segment), (scale bar = 2 μm, $n = 20$). (H–K) Morris Water Maze Test. Swimming speed (H), the latency to find the hidden platform from day 1 to day 5 was recorded ($P = 0.0304$) (I). Spatial memory was assessed on the 6th day by measuring the time spent in the target quadrant ($P = 0.0019$), (J) and the number of target platform crossings ($P = 0.0091$), (K) ($n = 10$). (L, M) Fear conditioning test was used to measure contextual memory. The freezing duration was measured during the 3-min memory test at 2 h ($P = 0.0013$), (L) and 24 h ($P = 0.0005$), (M) after conditioning, ($n = 10$). Hippocampal CA3-CA1 LTP (N) and its quantification ($P = 0.0330$), (O) were recorded by using the MED64 system ($n = 4$). Data were expressed as mean ± SD, A two-tailed Student's $T$ test was used for statistical analysis in (F–H, J–L, M, O). Two-way ANOVA of Sidak's multiple comparisons test was used for statistical analysis in (E, I). *$P < 0.05$, **$P < 0.01$, ***$P < 0.001$, ****$P < 0.0001$. vs Control group. Source data are available online for this figure.

synaptic and cognitive deficits associated with TGF-β1 in AngII-related hypertension.

## Conclusions

In summary, our study highlights a novel pathogenic link between AngII-related hypertension and cognitive impairments. This occurs as the result of increased TGF-β1 that upregulates the phosphorylation of Smad2/3, which then leads to the nuclear translocation of Smad complex may trigger the transcription of Sp1 and ultimately leading to synaptic damage and cognitive impairments (Fig. 8). Given the detrimental effects of the TGF-β1/Smad/Sp1 axis on synaptic function and cognition in AngII-related hypertension, targeting this pathway may offer a promising strategy for treating AngII-related hypertension associated cognitive dysfunction.

## Methods

**Reagents and tools table**

| Reagent/resource | Reference or source | Identifier or catalog number |
|---|---|---|
| **Experimental models** | | |
| Rat: SD rat | Animal Center of Tongji Medical College | |
| **Recombinant DNA** | | |
| Smad3 S423/425D | GENECHEM Bioscience | |
| Smad3 S423/425A | GENECHEM Bioscience | |
| **Antibodies** | | |
| β-actin | Proteintech | 66009-1-Ig |
| LaminB1 | abcam | ab16048 |
| TGF-β1 | abcam | ab92486 |
| Smad2/3 | Cell signal technology | 3102S |
| Phospho-Smad2 (Ser465/467)/Smad3 (Ser423/425) | Cell signal technology | 8828S |
| Sp1 | santa | sc-420 |
| C-Jun | Cell signal technology | #2315 |
| C-Fos | santa | sc-271243 |

| Reagent/resource | Reference or source | Identifier or catalog number |
|---|---|---|
| IRF-7 | santa | sc-74472 |
| HRP-conjugated Goat anti-Mouse IgG (H + L) | Abclonal | AS003 |
| HRP-conjugated Goat anti-Rabbit IgG (H + L) | Abclonal | AS014 |
| **Oligonucleotides and other sequence-based reagents** | | |
| primers for *HSP70* PCR | sense: TGCGTGGGCGTGTTCCA antisense: CGGTGTTCTGCGGGTTC | |
| primers for *BACE1* PCR | sense: CGGCGGGAGTGGTATTA antisense: AACGGTGCCTGTGGATG | |
| primers for *GAPDH* PCR | sense: TGCCTTCTCTTGTGACAAAGTGG antisense: CATTGCTGACAATCTTGAGGGAG | |
| **Chemicals, enzymes and other reagents** | | |
| AAV- CMV-TGF-β1-EGFP | GENECHEM Bioscience | |
| AAV- CMV-MCS-EGFP | GENECHEM Bioscience | |
| AAV-U6-Sh-TGF-β1-CAG-EGFP | GENECHEM Bioscience | |
| AAV-U6 -MCS-CAG-EGFP | GENECHEM Bioscience | |
| LV-U6-Sh-TGF-β1-CAG-EGFP | GENECHEM Bioscience | |
| LV-U6-MCS-CAG-EGFP | GENECHEM Bioscience | |
| LV-Ubi-Sp1-SV40-Cherry | GENECHEM Bioscience | |
| LV-Ubi-SV40-Cherry | GENECHEM Bioscience | |
| LV-U6-Sh-Sp1-Ubiquitin-Cherry | GENECHEM Bioscience | |
| LV-U6-MCS-Ubiquitin-Cherry | GENECHEM Bioscience | |
| implantable capsule micro-osmotic pump | ALZET | 2004 |
| Angiotensin II | Sigma | A9525 |
| SIS3 HCl | Selleckchem | S7959 |

| Reagent/resource | Reference or source | Identifier or catalog number |
|---|---|---|
| Lipofectamine 3000 | Thermo Fisher | L3000001 |
| Chip Kit | WUHAN GENECREATE BIOLOGICAL ENGINEERING | #JKR23002A |
| TGF-β1 ELISA kit | Wuhan Tanda Biotechnology | TD-D751002 |
| AngII ELISA kit | Wuhan Tanda Biotechnology | TD-D731188 |
| FD Rapid GolgiStain Kit | FD Neurotechnology | PK401 |
| DMEM/F12 | Gibco | 11965092 |
| Neurobasal | Gibco | 21103049 |
| B-27 supplement | Gibco | 17504044 |
| GlutaMAX Supplement | Gibco | 35050061 |
| **Software** | | |
| ImageJ | https://imagej.net/software/imagej/ | |
| GraphPad Prism 8 | https://www.graphpad.com/scientific-software/prism/ | |
| GEO Database | https://www.ncbi.nlm.nih.gov/geo/query/acc.cgi?acc | |
| **Other** | | |

## Animals

Two-month-old male SD rats, supplied by the Experimental Animal Center of Tongji Medical College, Huazhong University of Science and Technology, were housed with free access to food and water under a 12-hour light/dark cycle. All animal experiments received approval from the Institutional Animal Care and Use Committee (IACUC) of Huazhong University of Science and Technology (IACUC NO.4521).

In our study, we administered rats with AngII for 28 days using a subcutaneous pump to significantly increase blood pressure to induce hypertension. To further evaluate the role of TGF-β1, we conducted bilateral intra-hippocampal CA1 injections to rats with AAV- TGF-β1virus and explored the role of TGF-β1 in cognitive function. Besides, we consistently bilaterally administered rats with intra-hippocampal CA1 injections of either vector or AAV-Sh-TGF-β1 virus. Rats in the control group received vector virus injections into the hippocampal CA1 and allowed for a period of 4 weeks for expression. The Mod group was established by administering AngII via a subcutaneous tube for 28 days. The third group received simultaneous injections of AAV-Sh-TGF-β1 virus into the hippocampal CA1 region and a 28-day administration of AngII via a subcutaneous tube and designated as the AngII+Sh-TGF-β1 group.

To further evaluate the role of Sp1, we conducted bilateral intra-hippocampal CA1 injections to rats with LV-Sp1virus and explored the role of Sp1 in cognitive function. In addition, we conducted bilateral intra-hippocampal CA1 injections to rats with LV-TGF-β1 virus and LV-Sh-Sp1 virus. Rats in the control group received vector virus injections into the hippocampal CA1. TGF-β1 virus injecting CA1 was performed to establish the Mod group. The third group of rats received injections of both TGF-β1 and Sh-Sp1

viruses into the hippocampal CA1 region and was designated as the TGF-β1 + Sh-Sp1 group.

## Hypertension model

Rats were administered with AngII solution via a micro-osmotic pump (ALZET Osmotic Pump, size 2004). The AngII was continuously infused for 28 days at a rate of 600 ng/kg/min. Initially, the micro-osmotic pumps containing either AngII or saline were pre-soaked in 0.9% physiological saline at 37 °C for 24 h. The rats were anesthetized with isoflurane, and the back skin was depilated, and bluntly separated, and then the pumps were subcutaneously embedded and pre-filled with either AngII or saline before suturing.

## Blood pressure measurement

Non-invasive blood pressure measurements were conducted using a tail-cuff system. Choose a warm, quiet room for blood pressure measurement, the rat tail temperature should not be lower than 32 °C. Briefly, rats were placed in a restrainer on a heated platform for 10 min, after which the cuff was attached to the tail to record the blood pressure. The reported systolic and diastolic pressures are the averages of three consecutive measurements.

## NH3-PET brain imaging

Rats were anesthetized with a 2% isoflurane-oxygen mixture, and 13N-ammonia (13N-NH3) was administered intravenously in the tail at a dosage of 500 ± 10 μCi. Scanning was initiated immediately after the anesthesia and continued dynamically for 10 min. Images were reconstructed using the full 3D OSEM (Ordered Subset Expectation Maximization) method. Blood flow in the hippocampus and the brain was assessed using 13N-ammonia uptake.

## Stereotaxic surgery

Rats were anesthetized with isoflurane, before being positioned in a stereotactic apparatus. Following disinfection with iodophor and 75% alcohol, the scalp was incised along the midline between the ears. Stereotaxic drilling was performed bilaterally in the skull at coordinates of 4.2 mm posterior, 3 mm lateral, and 2.8 mm ventral relative to the bregma using a microinjection system (World Precision Instruments). The virus was injected into the hippocampal region at a rate of 0.125 μL/min. Following a 10- or 15-min infusion, the needle was gradually withdrawn, and the incision was sutured. After lateral ventricle stereotactic surgery, the rats were placed on a temperature-controlled pad to recover. After virus injection for 4 weeks, behavioral tests were performed.

## Behavior tests

### Open field

The open field test is a behavioral assay designed to assess autonomic motor activity and anxiety-related behaviors. The apparatus consists of a standard open field: a $100 \times 100$ cm$^2$ square arena made of PVC with walls 70 cm high. Rats were placed at the same starting point and allowed to explore the arena freely for 5 min individually. After each rat completed the test, the arena was

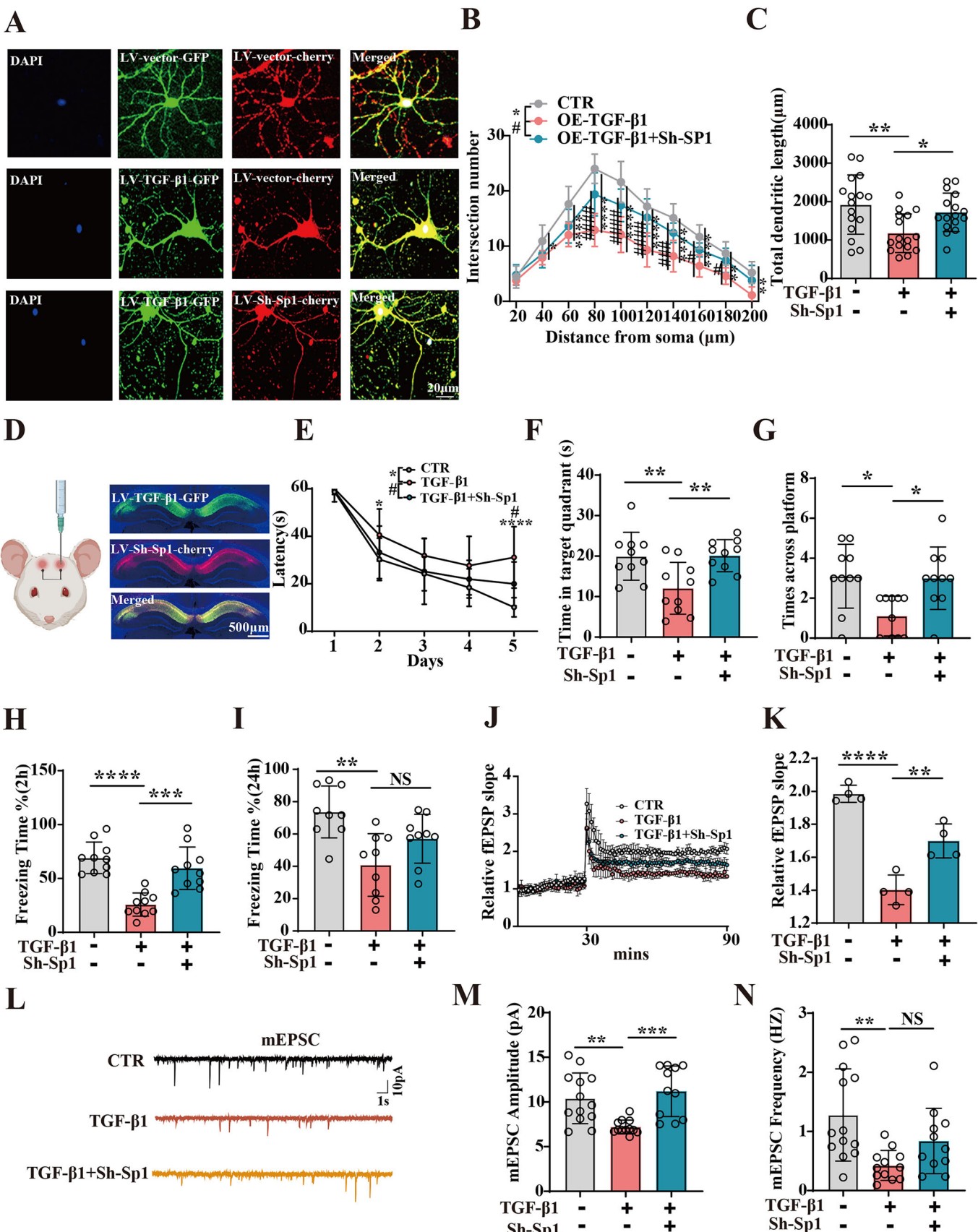

**Figure 7.   Downregulation of Sp1 improved TGF-β1-induced synaptic and cognitive deficits.**

(A) Hippocampal primary neurons were administered with LV-TGF-β1 or Sh-Sp1 virus. (B) Sholl analysis (scale bar = 20 μm, $n$ = 10), (Row4 $P$ = 0.0259; Row6 $P$ < 0.0001; Row8 $P$ < 0.0001, $P$ < 0.0001; Row10 $P$ < 0.0001, $P$ < 0.0001; Row12 $P$ < 0.0001, $P$ < 0.0001; Row14 $P$ < 0.0001, $P$ = 0.0009; Row16 $P$ < 0.0001, $P$ = 0.0327; Row18 $P$ = 0.0012, $P$ = 0.0410; Row20 $P$ = 0.0012). (C) Quantitative analyses of dendritic length ($P$ = 0.0031, $P$ = 0.0382), ($n$ = 16). (D) OE-TGF-β1 virus or Sh-Sp1 was injected into the hippocampal CA1 region of the SD rats, and GFP and mCherry expression were observed 4 weeks after injection. (E–G) Morris Water Maze Test. The latency to find the hidden platform (E) from day 1 to day 5 was recorded, (Row2, $P$ = 0.0293; Row5 $P$ < 0.0001, $P$ = 0.0160), ($n$ = 10). Spatial memory was assessed on the 6th day by measuring the time spent in the target quadrant ($P$ = 0.0096, $P$ = 0.0083), ($n$ = 10), (F) and the number of target platform crossings ($P$ = 0.0103, $P$ = 0.0150), ($n$ = 10), (G). (H, I) Fear conditioning test was used to measure contextual memory. The freezing duration was measured during the 3-min memory test at 2 h ($P$ < 0.0001, $P$ = 0.0001), ($n$ = 10), (H) and 24 h ($P$ = 0.0011), ($n$ = 9), (I) after conditioning. (J) Hippocampal CA3-CA1 LTP and its quantification (K) were recorded by using the MED64 system ($P$ < 0.0001, $P$ = 0.0021), ($n$ = 4). (L) Rat hippocampal CA1 neurons were subjected to whole-cell patch-clamp recording of mEPSCs, the amplitude ($P$ = 0.0048, $P$ = 0.0008), (M) and frequency ($P$ = 0.0023), (N) of mEPSCs were measured ($n$ = 11–13 recordings/4 rats per group). Data were expressed as mean ± SD, one-way ANOVA with Tukey's multiple comparisons test was used for statistical analysis in (C, F, G, H, I, K, M, N). Two-way ANOVA with Tukey's multiple comparisons test was used for statistical analysis in (B, E). *$P$ < 0.05, #$P$ < 0.05, **$P$ < 0.01, ***$P$ < 0.001, ###$P$ < 0.001, ####$P$ < 0.0001, ****$P$ < 0.0001. vs TGF-β1 group. Source data are available online for this figure.

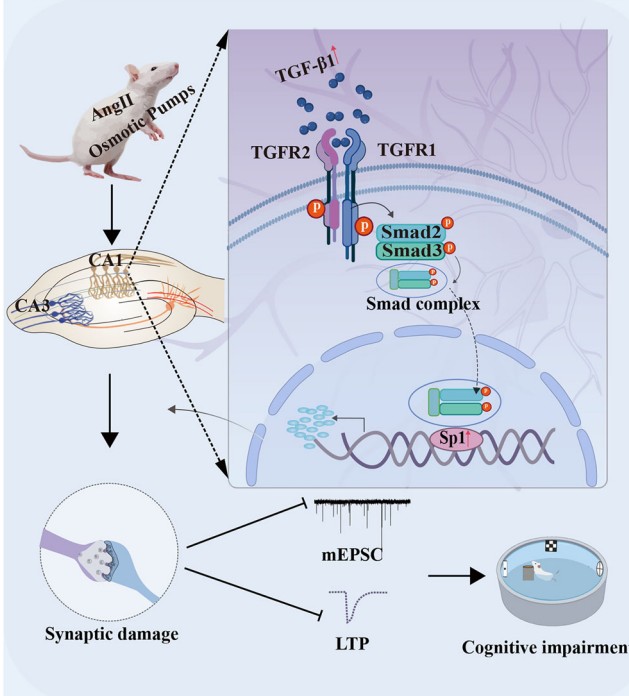

**Figure 8.   Schematic illustration of the TGF-β1/SMAD2/3/Sp1 axis exerts synaptic damage associated with AngII-related hypertension and cognitive impairments.**

AngII-related hypertension upregulates TGF-β1 binding to TGF-β1 receptors triggering increased Smad2/3 phosphorylation. These results in the translocation of the Smad complex into the nucleus and its binding to the transcription factor Sp1, thus leading to synaptic damage and cognitive impairments.

apparatus consists of two symmetrical open arms without walls and two symmetrical closed arms with walls, connected by a central platform. Each rat was allowed to explore the apparatus freely for 5 min. Rat was allowed to freely explore the arena for 5 min. Between trials, the walls and floor were cleaned with diluted ethanol. The number of entries into the open and closed arms, as well as the sedentary time, were recorded.

## Morris water maze test

The Morris water maze test was conducted to assess hippocampus-dependent spatial learning and memory. The test was conducted according to the methods described in the reference (Guo et al, 2020). A circular pool filled with water was used, with a platform submerged 1 cm below the water surface. Black ink was added to the water to obscure the submerged platform. Geometric shapes were affixed to the walls as spatial reference cues. During the acquisition phase, rats were trained to locate the platform within 60 s across four quadrants. If the rat failed to locate the platform within 60 s, the escape latency was recorded as 60 s. The rat was then guided to remain on the platform for 20 s to familiarize itself with the platform's orientation. The test was conducted over five consecutive days, with four trials per day starting from different quadrants. On the sixth day, the platform was removed, and a probe test was conducted to evaluate spatial memory retention. The latency to reach the platform, swimming speed, time spent in the target quadrant, and the number of platform crossings were automatically monitored and recorded using a video tracking system.

## Fear conditioning test

Fear conditioning test was conducted to assess episodic memory. On the first day, rats were placed in the chamber for 3 min before exposure to a sound stimulus (70 dB for 30 s) which will immediately be followed by a brief electrical shock (0.8 mA for 3 s). This procedure was repeated twice, with each passage spaced by a 2-minute interval. Two hours later, rats were returned to the same chamber for 3 min without any stimuli, and their freezing beha conclusion vior was recorded to assess memory retention. On the following day, the rats were again placed in the same chamber without any stimuli for 3 min, and freezing time was recorded to evaluate memory retention.

cleaned with diluted ethanol to eliminate odor cues, ensuring a pristine environment for subsequent behavioral experiments. The times across central square of the open field and the total distance traveled were automatically tracked and recorded.

## Elevated plus maze test

The elevated plus maze (EPM) is used to assess the anxiety state of rats by exploiting their conflicting tendencies to explore novel environments and to avoid the exposed open arms. The EPM

## LTP recording

Rats were anesthetized, and their brains were removed and immediately placed in ice-cold artificial cerebrospinal fluid (aCSF), brain slices were prepared as previously described (Guo et al, 2020). For LTP recordings, slices were placed in a recording chamber equipped with an $8 \times 8$ microelectrode array at the base. Signal acquisition was performed using the MED64 System by Alpha MED Sciences (Panasonic). Field excitatory postsynaptic potentials (fEPSPs) in CA1 neurons were elicited by stimulating CA3 neurons. LTP was induced using three trains of high-frequency stimulation (HFS; 100 Hz, 1 s each). The fEPSP slope was quantified over 60-min.

## Electrophysiological recordings

Neurons of the hippocampus CA1 region were visualized for whole cell recording and the pipette resistance was in the range of 3–5 M$\Omega$. The intracellular solution contained (in mM): 128 K-gluconate, 5 MgATP, 1.1 EGTA, 10 HEPES, 0.4 Na2GTP, 10 Naphosphocreatine, pH 7.2–7.4, 280–290 mOsm. 5 mM QX-314 was added to block voltage-gated Na+ channels and GABABRs. The mEPSC needs to be clamped at $-60$ mV and in the presence of 5 μM tetrodotoxin (TTX) and 20 μM picrotoxin (PTX). The internal solution to record IPSC contained (in mM): 140 CsCl, 10 HEPES, 2 MgCl$_2$, 0.5 EGTA, 2 MgATP, 0.5 Na 3GTP, 12 phosphocreatine, pH 7.2–7.4 and 280–290 mOsm. When recording mIPSC, the potential should be clamped at $-60$ mV. 5 μM TTX and 50 μM kynurenic acid (KYN) should be added. Whole-cell recordings were performed using a Multiclamp 700B amplifier. Data was analyzed by pClamp 10.0 (Molecular Devices). The whole-cell currents were filtered at 5 kHz with a low-pass Bessel filter and digitized between 5 and 20 kHz.

Paired pulse ratios were calculated as a ratio of EPSC2 to EPSC1 separated by inter-stimulus intervals of 20, 50, 100 and 200 ms. To measure the input-output curves for pyramidal neurons, a bipolar tungsten electrode was placed to the stratum radiatum with a bipolar tungsten electrode ~50 μm from the recording electrode to test potential presynaptic effects.

## Primary hippocampal neuron culture

Primary hippocampal neurons were isolated from an embryonic day 18 (E18) Sprague-Dawley rats. Hippocampal tissues were dissected, gently minced in Hank's Buffered Saline Solution, and then incubated in 0.25% (vol/vol) trypsin at 37 °C for 12 min. Neurons were seeded in culture dishes pre-coated with 100 μg/mL poly-D-lysine and maintained in a Neurobasal medium supplemented with 2% B-27 and 1× GlutaMAX, suitable for subsequent viral and pharmacological interventions. Following treatments with the experimental requirements by using SIS$_3$, LV-TGF-β1/LV-Sp1 and LV-Sh-Sp1, cells were either lysed in RIPA buffer for biochemical analyses or fixed with 4% paraformaldehyde for imaging.

## Western blotting and co-immunoprecipitation (Co-IP)

Hippocampal tissue or primary hippocampal neurons were homogenized in RIPA buffer and then centrifuged at 12,000 rpm for 10 min. The supernatant was collected for protein analysis. Nuclear fractions were isolated using the NE-PER Nuclear and Cytoplasmic Extraction Kit (Pierce), according to the manufacturer's instructions. Proteins in the extracts were separated using SDS-PAGE. The imprints were then coupled to HRP-conjugated Goat anti-mouse/anti-rabbit IgG at 25 °C for 1 h, and the images were viewed with the ChemiScope (Clinx Science Instruments Co. Ltd.). Co-IP experiment was used to analyze protein interactions. The Smad2/3 antibody was incubated with the samples overnight at 4 °C, then washed with PBS and boiled at 95 °C for 10 min, the target proteins were analyzed by western blotting.

## Chromatin immunoprecipitation (Chip) assay

The N2A cells were maintained in a humidified incubator at 37 °C with 5% CO$_2$. For plasmid transfection, cells were seeded into a six-well plate and incubated for 24 h prior to transient transfection. A mixture containing 3 μg of Smad3 S423/425D (Phosphorylation-activated mutation), S423/425 A (Nonphosphorylated mutation) plasmid or vector and 3 μl of Lipofectamine 3000 (Thermo Fisher, USA) was used for transfection in each well of the 6-well plates. Chip was performed using the Chip Kit (Catalog #JKR23002A, WUHAN GENECREATE BIOLOGICAL ENGINEERING CO., LTD.). For immunoprecipitation, 3 μg of target Sp1 antibody (IP group) or 1 μL of the same species IgG (IgG group) was added and incubated overnight at 4 °C. Elution and Chip-qPCR Procedure: For Chip-qPCR, 2 μL of each sample was added into the PCR reaction wells, with duplicate wells for each sample. The remaining components were prepared according to the SYBR Green qPCR Mix instructions. After mixing, the samples were labeled, briefly centrifuged for 10 s, and loaded into the fluorescence quantitative PCR system for analysis. The primers for *HSP70* sense: TGCGTGGGCGTGTTCCA antisense: CGGTGTTCTGCGGGTTC. *BACE1* sense: CGGCGGGAGTGGTAT TA antisense: AACGGTGCCTGTGGATG, and *GAPDH* sense: TGCC TTCTCTTGTGACAAAGTGG antisense: CATTGCTGACAATCTT GAGGGAG are listed.

## Enzyme-linked immunosorbent (ELISA)

Hippocampal tissues and plasma were lysed in PBS buffer (0.01 M, pH 7.4), centrifuged at 1500–5000 × $g$ for 10–30 min at 4 °C, and the supernatant was collected. Anti-rat AngII and TGF-β1 ELISA kits were employed to quantify the levels of AngII and TGF-β1, according to the manufacturer's instructions. The rat AngII and TGF-β1 ELISA kits were purchased from Wuhan Tanda Biotechnology (Product Nos. TD-D731188, TD-D751002).

## Golgi staining

Rats were anesthetized using isoflurane and subsequently perfused with saline. Following perfusion, the entire brain was extracted and immersed in a Golgi solution for one month. Utilizing the FD Rapid Golgi Stain Kit (PK401, FD Neurotechnology) following the manufacturer's guidelines. The brain was sectioned into 80 μm thick slices using a vibrating blade slicer (Leica VT1000S, Germany). Images were acquired using a microscope (Nikon, Tokyo, Japan) and analyzed as previously described (Hoover et al, 2010; Yin et al, 2016). Sholl analysis (using the cell body as the center of concentric circles and counting the intersection points

with circles of different radii to reflect the number of dendritic branches). Dendritic length was conducted using Image software.

## Datasets collection and processing

In this study, bioinformatics analyses were conducted to identify marker genes associated with Angiotensin II (AngII) models and the function of Sp1. The datasets GSE47529 (Azevedo et al, 2014), GSE126500 (Gilmour et al, 2019), GSE165771 (Ni et al, 2024), GSE37935 (Oleaga et al, 2012), and GSE31628 (Felthaus et al, 2012) were obtained from the Gene Expression Omnibus (GEO) database. Corresponding mRNA expression matrices were aligned for comprehensive analysis.

In the GSE47529 dataset, C6 rat glioma cells were treated with or without AngII. The control group consisted of cells from GSM1151627-GSM1151630 and GSM1151643-GSM1151646, serving as the reference for bioinformatics analysis. The experimental group, representing the AngII models, included cells from GSM1151631-GSM1151634 and GSM1151647-GSM1151650, which were exposed to AngII for varying durations. The mRNA expression matrices from these samples were normalized using the "limma" R package. Outliers, specifically GSM1151632 and GSM1151628, were removed using the "cutreeStatic" function in the "WGCNA" R package.

Additionally, several datasets were utilized to investigate the function of Sp1. The GSE165771 dataset included HEK293 cells following Sp1 knockdown. The GSE37935 dataset comprised HeLa cells treated with siRNA targeting Sp1 mRNA (siSp1). The GSE31628 dataset contained SHED cells overexpressing Sp1. The GSE126500 dataset featured ESC, Flk, HE1, HE2, and progenitor cells lacking Sp1 DNA binding activity. mRNA data from GSE126500 were converted into human homologs and normalized using the "limma" R package.

These analyses were aimed at elucidating the roles of AngII and Sp1 in cellular function and identifying potential biomarker genes for further research.

## Bioinformatics analysis

For the dataset GSE47529, Differential Expression Genes (DEGs) were identified using the "limma" R package with criteria set at | Log2FoldChange | >0.35 and $P < 0.05$. Gene Ontology (GO) enrichment analysis and Kyoto Encyclopedia of Genes and Genomes (KEGG) pathway enrichment analysis were conducted using the "biomaRt" and "clusterProfiler" R packages, with a significance threshold of $P$ value < 0.05 for the enrichment terms. Visualization of KEGG and GO enrichment analysis results were performed using the "ggsankey", "aPEAR", "GOplot", and "ggplot2" R packages. Venn diagrams illustrated the intersection of DEGs with core genes in the "Neuroactive ligand-receptor interaction" and "Cellular senescence" pathways. Protein–Protein Interaction (PPI) networks were constructed using the brain network (edge weight >0.5) from the TissueNexus online database. The top 5 hub genes were identified through EcCentricity analysis using the Cytohubba plug-in in Cytoscape 3.10.2 software. For the datasets GSE126500, GSE165771, GSE37935, and GSE31628, samples were categorized into groups based on Sp1 expression levels (low vs. high). DEGs were identified using the "limma" R package with a $P < 0.05$ criterion. Venn diagrams were employed to visualize the intersection of these DEGs (65 genes). The top 10 hub genes were identified

through MCC analysis using the Cytohubba plug-in in Cytoscape 3.10.2 software.

## Statistical analysis

In vitro assays, the experiments were conducted by a blind individual, who did not know the sample information. The animal's information for vivo assays was decoded after all samples were analyzed. Sample size was determined by Power and Precision (Biostat). Data are expressed as mean ± SD and analyzed using GraphPad Prism statistical software (GraphPad Software). The one-way ANOVA was used to determine the differences among groups. For the comparison between the two groups, the Student's $t$ test was used. The significance was assessed at $P < 0.05$. All results shown correspond to individual representative experiments.

## Ethics approval and consent to participate

No humans were used in this research. All animal experiments received approval from the Institutional Animal Care and Use Committee (IACUC) of Huazhong University of Science and Technology (IACUC No. 4521).

## Data availability

This study includes the URL for the GEO deposition: https://www.ncbi.nlm.nih.gov/geo/query/acc.cgi?acc=GSE47529 https://www.ncbi.nlm.nih.gov/geo/query/acc.cgi?acc=GSE126500 https://www.ncbi.nlm.nih.gov/geo/query/acc.cgi?acc=GSE165771 https://www.ncbi.nlm.nih.gov/geo/query/acc.cgi?acc=GSE37935 https://www.ncbi.nlm.nih.gov/geo/query/acc.cgi?acc=GSE31628.

The source data of this paper are collected in the following database record: biostudies:S-SCDT-10_1038-S44319-025-00470-0.

## Peer review information

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

## Acknowledgements

We thank the Medical Subcenter of HUST Analytical & Testing Center for data acquisition. The authors are grateful to Mr. Dan Ke and Ms. Qun Wang for helpful technical suggestions during the conduct of this study. This work is supported in parts by grants from National Natural Science Foundation of China (No. 82330041, 82360263), grant from Science and Technology Innovation Team project to Xiaochuan Wang from the Department of Science and Technology of Hubei Province (No. 2022-72-18), Science, Technology and Innovation Commission of Shenzhen Municipality (JCYJ20210324141405014, JCYJ20220530161207016) and grant from the Central Government Guides Local Science and Technology Development Special Projects of Hubei Province (No. 2022BGE243).

## Author contributions

**Cuiping Guo**: Conceptualization; Resources; Data curation; Formal analysis; Supervision; Writing—original draft. **Wensheng Li**: Software; Methodology. **Yuanyuan Li**: Data curation; Formal analysis. **Yi Liu**: Methodology; Writing—review and editing. **Yacoubou Abdoul Razak Mahaman**: Writing—review and editing. **Jianzhi Wang**: Investigation; Writing—review and editing. **Hongbin Luo**: Funding acquisition; Writing—review and editing. **Rong Liu**: Writing—review and editing. **Hui Shen**: Conceptualization; Data curation; Investigation; Project administration; Writing—review and editing. **Xiaochuan Wang**: Conceptualization; Resources; Supervision; Funding acquisition; Writing—review and editing.

Source data underlying figure panels in this paper may have individual authorship assigned. Where available, figure panel/source data authorship is listed in the following database record: biostudies:S-SCDT-10_1038-S44319-025-00470-0.

## Disclosure and competing interests statement

The authors declare no competing interests.

# Expanded View Figures

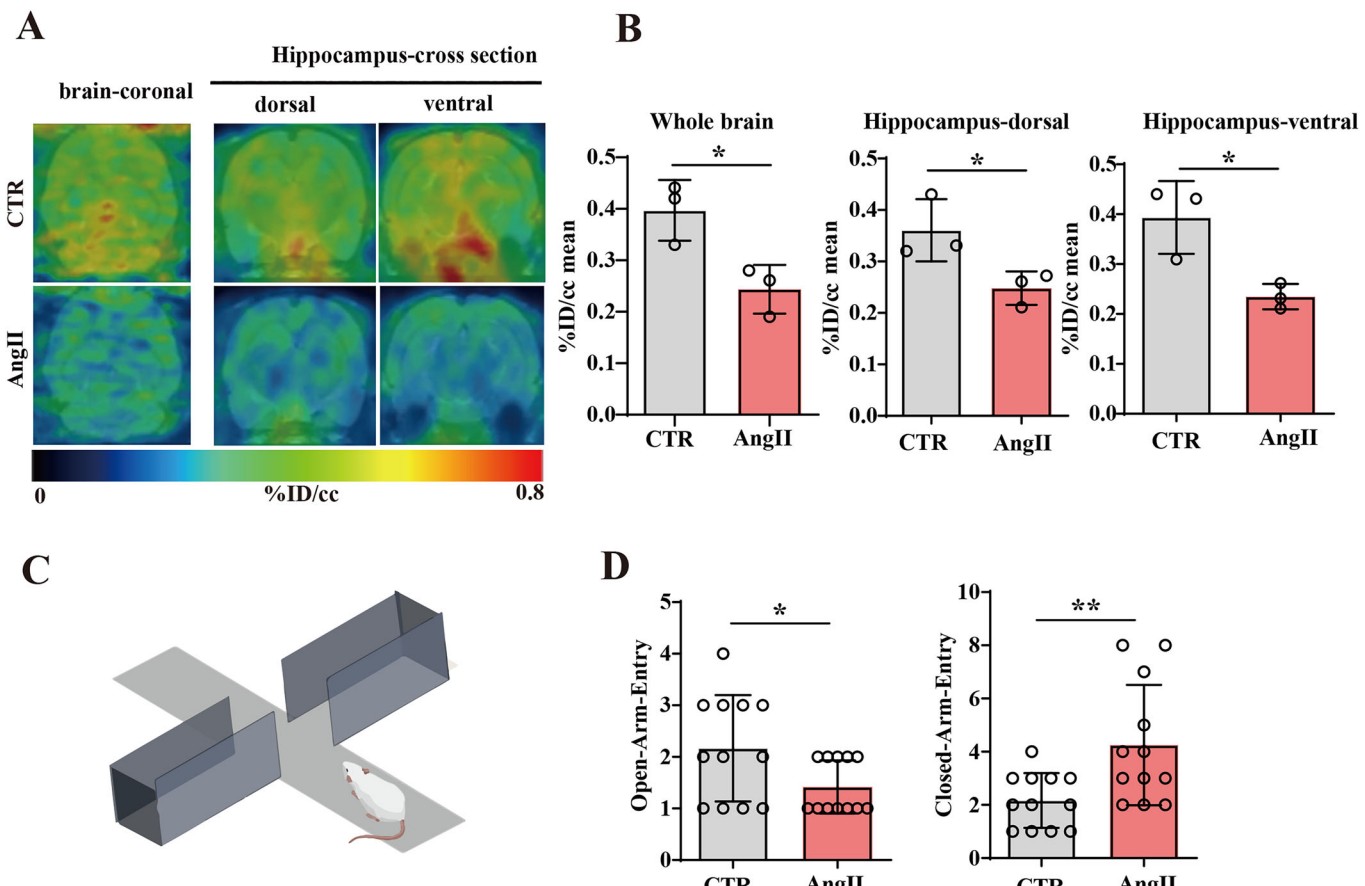

**Figure EV1. AngII-related hypertension reduced blood flow and cognitive impairments.**

(A) The 13N-NH3-PET rat brain dynamic imaging was used to observe the blood flow in the brain and hippocampus. (B) The results showed that the whole brain ($P = 0.0246$), the hippocampus dorsal ($P = 0.0472$) and ventral ($P = 0.0236$), ($n = 3$). (C, D) The high plus maze test was performed to measure the entries to the open ($P = 0.0343$) and closed arms ($P = 0.0082$), ($n = 12$). Data are presented as Mean ± SD. A two-tailed Student's *T* test was used for statistical analysis in (B, D). *$P < 0.05$, **$P < 0.01$, versus Control group. Source data are available online for this figure.

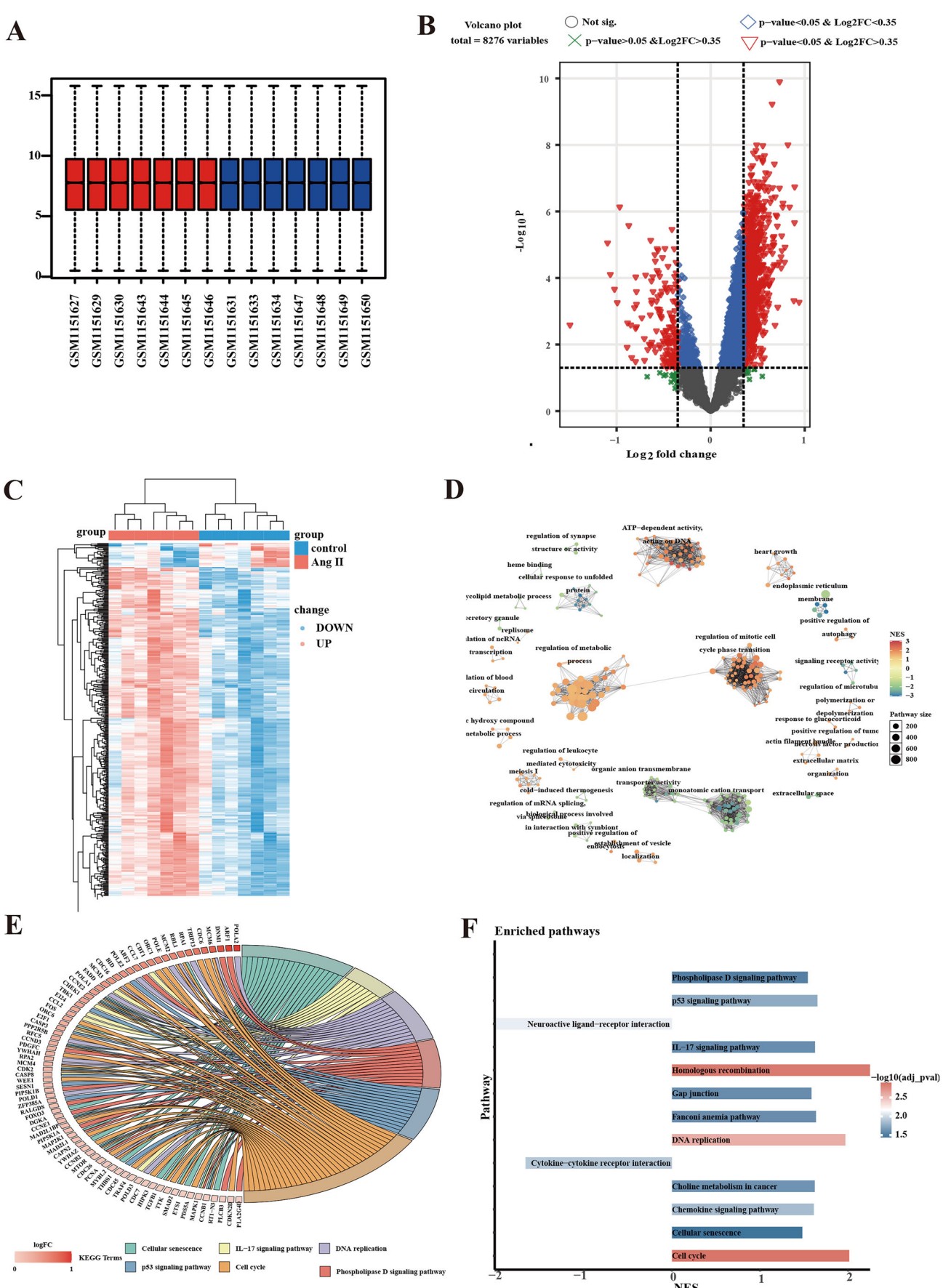

◀ **Figure EV2.  The visualization of GSE47529 gene expression analysis results.**

(**A**) The boxplots of normalized gene expression. (**B**) A volcano plot of differentially expressed genes, ($n = 7$). (**C**) DEG expression heat map of differentially expressed genes. (**D**) Enrichment network displaying the GO enrichment analysis results for GSE47529. (**E**) Visualization of selected KEGG pathways and their core enrichment genes in GSE47529. (**F**) Top 13 KEGG pathways enriched in GSE47529 DEGs.

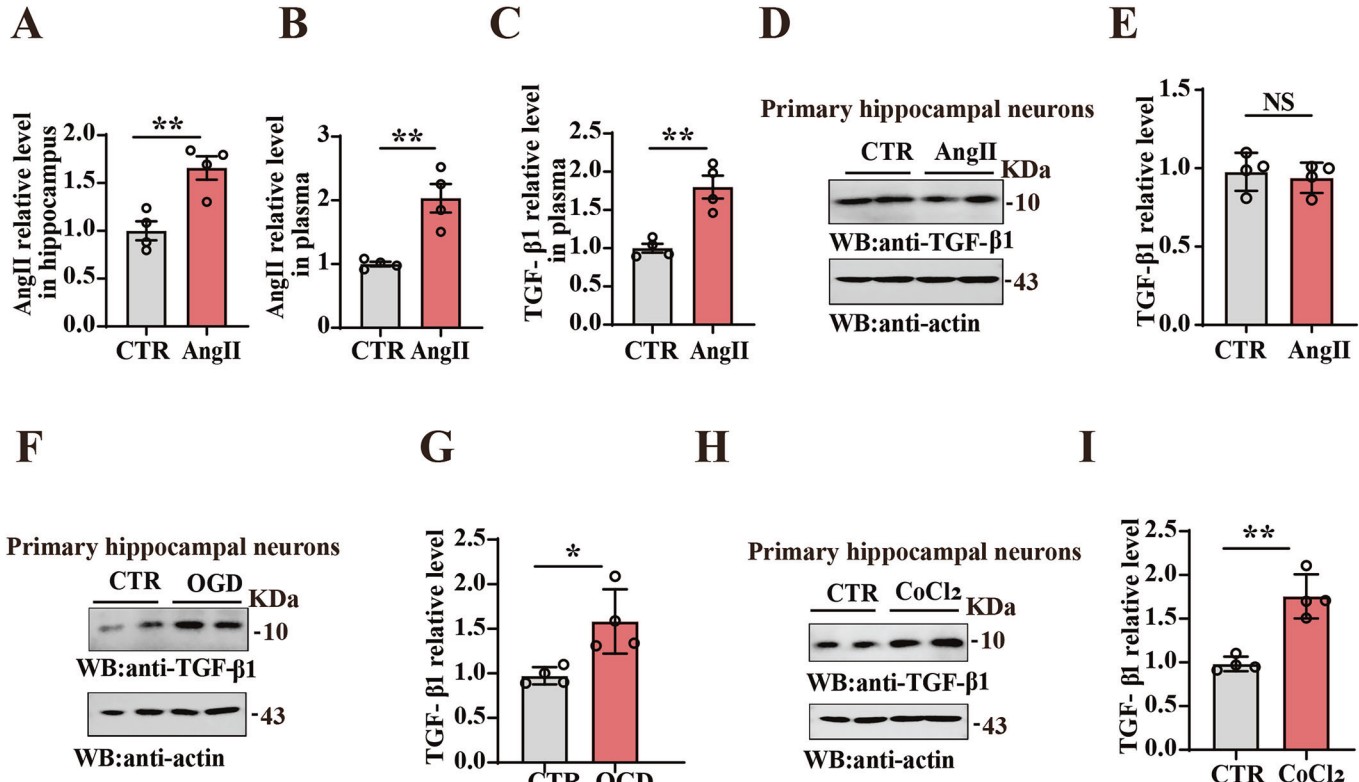

**Figure EV3. The AngII and TGF-β1 levels were tested with different conditions.**

The AngII levels were measured in hippocampus ($P = 0.0059$), ($n = 4$), (**A**) and in plasma ($P = 0.0038$), ($n = 4$), (**B**) in the two groups ($n = 4$). The TGF-β1 levels were measured in plasma ($P = 0.0025$), ($n = 4$), (**C**) in the two groups. (**D**) Hippocampal primary neurons were treated with compound AngII. Western blotting showed the TGF-β1 levels after treatment of hippocampal neurons with AngII 0.5 μM. Actin was used as a loading control. (**E**) Quantitative analysis of the TGF-β1 levels ($n = 4$). (**F**) Hippocampal primary neurons were treated with oxygen-glucose deprivation (OGD) for 12 h. Western blotting showed the TGF-β1 levels after OGD. Actin was used as a loading control. (**G**) Quantitative analysis of the TGF-β1 levels ($P = 0.0173$), ($n = 4$). (**H**) Hippocampal primary neurons were treated with CoCl$_2$ 100 μM for 6 h (CoCl$_2$ was a drug for chemical hypoxia model). Western blotting showed the TGF-β1 levels after CoCl$_2$. Actin was used as a loading control. (**I**) Quantitative analysis of the TGF-β1 levels ($P = 0.0011$), ($n = 4$). Data are presented as mean ± SD. A two-tailed Student's $T$ test was used for statistical analysis in (**A–C, E, G, I**). *$P < 0.05$, **$P < 0.01$, versus Control group. Source data are available online for this figure.

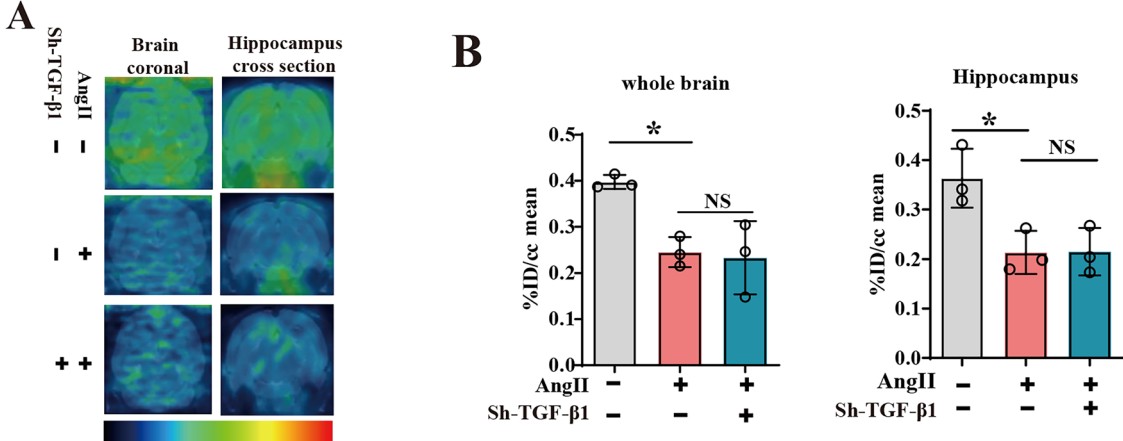

**Figure EV4.  Downregulation of TGF-β1 in AngII rats didn't change blood flow.**

(A) The 13N-NH3-PET rat brain dynamic imaging was used to observe the blood flow in the brain and hippocampus: the whole brain ($P = 0.0122$) and the hippocampus ($P = 0.0256$) were measured, ($n = 3$). (B) Data are presented as mean ± SD. One-way ANOVA with Tukey's multiple comparisons test was used for statistical analysis in (B). *$P < 0.05$, versus AngII group. Source data are available online for this figure.

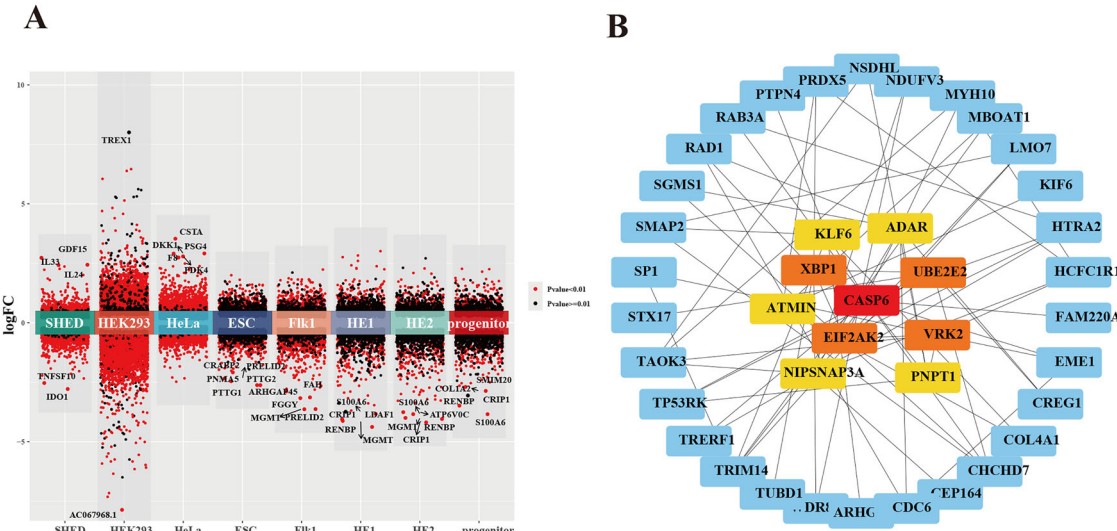

**Figure EV5. The visualization of GSE126500, GSE165771, GSE37935, and GSE31628 gene expression analysis results.**

(A) Scatter plot of Log2FoldChange (low Sp1 expression vs. high Sp1 expression) in different cell types, including SHED, HeLa, HEK293, ESC, Flk, HE1, HE2, and progenitor cells. (B) Protein–Protein Interaction (PPI) network and top 10 hub genes of the intersected genes.

