## [Peer Review File · EMBO Reports]

Elevated TGF- β 1 impairs synaptic and cognitive function through activation of Smad2/3-Sp1 pathway in AngII-related hypertension

Xiaochuan wang, Cuiping Guo, Wensheng Li, Yuanyuan Li, Yi Liu, Yacoubou Mahaman, Jianzhi wang, Hongbin Luo, Rong Liu, and Hui Shen

Corresponding author(s): Xiaochuan wang (wxch@mails.tjmu.edu.cn) , Hui Shen (shenhui@tmu.edu.cn)

Review Timeline:

Submission Date:	21st Aug 24
Editorial Decision:	13th Dec 24
Revision Received:	24th Jan 25
Editorial Decision:	31st Mar 25
Revision Received:	8th Apr 25
Accepted:	18th Apr 25

Editor: Bernd Pulverer

Transaction Report:

Dear Dr. Wang

Thank you for the submission of your manuscript entitled 'Elevated TGF- β 1 impairs cognitive function by activating smad2/3-Sp1 in AngII-related hypertension' to EMBO Reports. I would like to apologize for the terribly slow process in this case. I admit that we struggle to find a suitable panel of expert referees in this case. Thank you for your notable patience in awaiting this decision.

I am glad to say that the referees are in principle favourably inclined on your work, but you will see that the current dataset is judged to be premature and we agree with the need for the requested further developments.

Briefly, we appreciate that you provide evidence for an angiotensin II > TGF β > Smad/SP1 pathway that leads to hypertension and vascular dementia in rat. Angiotensin II treatment increases blood pressure, reduces hippocampal blood flow, leads to synaptic dysfunction and cognitive deficits, and is associated with elevated TGF- β 1. TGF- β 1 overexpression exacerbates cognitive impairment, while loss of function partially rescues the Ang II linked pathology. You show the effect is dependent on the canonical TGF- β 1 pathway i.e. Smad2/3, as well as SP1.

Both referees acknowledge that the findings are potentially interesting. Ref 1 notes 'the study was rigorously designed and well executed with solid data that supports the conclusion', and ref 2 'the observations are intriguing and may uncover the pathogenesis of Ang-II-related cognitive impairment'. However, ref 2 requires:

1) measure the amount of Ang-II in the brain and CSF.

2) definitive additional experimental evidence to test whether AngII regulates TGF- β 1 directly or indirectly (and, if that is the case, how; for example by hypoperfusion and decreased energy / oxygen supply).

3) measure peripheral (plasma) Ang-II and TGF- β 1 and demonstrate or exclude effects of TGF- β 1 pathway activation in the peripheral organs on the described neuronal effects.

4) ref 1 requires data to directly show TGF- β 1 expression in the CNS.

5) While not raised directly by the referees, I strongly encourage you to explore the mechanistic connection between Smad2/3 and SP1 in the context of transcriptional regulation as you suggest. Do these factors bind to relevant gene promoters together (we recommend ChIP or EMSAs if any relevant target genes are known). It is unclear what you mean with the statement '...promotes the entry of the phosphorylated smad2/3 complex into the nucleus, where they bind to Sp1 instead of the other transcription factors IRF-7, c-Jun, c-Fos.'. Please elaborate by adding experimental evidence that Smad2/3 phosphorylation regulates nuclear entry as claimed.

6) the referees also request more details to describe both the behavioral and intra-hippocampal injection experiments. Please ensure the methods section is sufficiently detailed as to render the experiments easily reproducible (we also encourage the link to protocols on e.g. protocols.io)

7) Please carefully revise the text with a view on optimizing the English and correcting typos e.g. in the abstract 'hypofusion'; in the conclusion, it is an overstatement (based on the current dataset) to claim '...triggering the transcription of Sp1 and ultimately leading to synaptic damage and cognitive impairments' > please provide support for this or revise the text.

Please simplify language where possible e.g. SP1 is a well known 'transcription factor' - considerably shorter than 'Specificity protein 1 (Sp1) is a widely DNA-binding protein expressed that both activates and inhibits gene transcription').

I would thus like to invite you to revise your manuscript with the understanding that all of the referee and editorial concerns must be fully addressed experimentally as outlined above.

Please address all referee concerns in a complete point-by-point response. Acceptance of the manuscript will depend on a positive outcome of a second round of review. It is EMBO reports policy to allow a single round of major revision only and acceptance or rejection of the manuscript will therefore depend on the completeness of your responses included in the next, final version of the manuscript.

We realize that it is difficult to revise to a specific deadline. In the interest of protecting the conceptual advance provided by the work, we recommend a revision within 3 months (9th Mar 2025). However, please discuss the revision progress ahead of this time with myself if you require more time to complete the revisions and we will review experimental progress and the literature at the time to decide on next steps.

1) A data availability section providing access to data deposited in public databases is missing. If you have not deposited any data, please add a sentence to the data availability section that explains that.

2) Your manuscript contains statistics and error bars based on n=2. Please use scatter plots in these cases. No statistics should be calculated if n=2.

3) We replaced Supplementary Information with Expanded View (EV) Figures and Tables that are collapsible/expandable online. A maximum of 5 EV Figures can be typeset. EV Figures should be cited as 'Figure EV1, Figure EV2' etc... in the text and their respective legends should be included in the main text after the legends of regular figures.

5) a complete author checklist, which you can download from our author guidelines <https://www.embopress.org/page/journal/14693178/authorguide>. Please insert information in the checklist that is also reflected in the manuscript. The completed author checklist will also be part of the RPF.

6) Please note that all corresponding authors are required to supply an ORCID ID for their name upon submission of a revised manuscript (<https://orcid.org/>). Please find instructions on how to link your ORCID ID to your account in our manuscript tracking system in our Author guidelines <https://www.embopress.org/page/journal/14693178/authorguide#authorshipguidelines>

- the name of the statistical test used to generate error bars and P values,
- the number (n) of independent experiments (please specify technical or biological replicates) underlying each data point,
- the nature of the bars and error bars (s.d., s.e.m.),
- If the data are obtained from n Program fragment delivered error ``Can't locate object method "less" via package "than" (perhaps you forgot to load "than"?) at //ejpvfs23/sites23b/embo/www/letters/embo_decision_revise_and_review.txt line 56.' 2, use scatter blots showing the individual data points.

12) All Materials and Methods need to be described in the main text using our 'Structured Methods' format, which is required for all research articles. According to this format, the Methods section includes a Reagents and Tools Table (listing key reagents, experimental models, software and relevant equipment and including their sources and relevant identifiers) followed by a Methods and Protocols section describing the methods using a step-by-step protocol format. The aim is to facilitate adoption of the methodologies across labs. More information on how to adhere to this format as well as a downloadable template (.docx) for the Reagents and Tools Table can be found in our author guidelines: <https://www.embopress.org/page/journal/14693178/authorguide#structuredmethods>.

An example of a Method paper with Structured Methods can be found here: <https://www.embopress.org/doi/full/10.1038/s44320-024-00037-6#sec-4>

I look forward to seeing a revised form of your manuscript when it is ready.

Yours sincerely,

~~~~~  
Bernd Pulverer, Ph.D.  
Chief Editor, EMBO Reports  
EMBO  
Meyerhofstrasse 1, D-69117 Heidelberg  
Tel: +4962218891501  
[bernd.pulverer@embo.org](mailto:bernd.pulverer@embo.org)  
~~~~~

Referee #1:

Guo and colleagues established an animal model where AngII-induced hypertension caused synaptic and cognitive deficits in rats. They found TGFbeta1 expression phenocopying AngII treatment in terms of synaptic and cognitive deficits and inhibition of TGFbeta1 in AngII-treated rats rescued cognitive deficits. They further demonstrated that this was likely mediated by Smad2/3-Sp1 signaling. They concluded that TGFbeta1/Smad2/3/Sp1 axis plays a critical role in mediating AngII-induced cognitive deficits. Overall, the study was rigorously designed and well executed with solid data that supports the conclusion. There were several minor concerns:

- 1) While the authors demonstrated GFP expression in the hippocampus after virus injection, it was not confirmed whether TGF1beta1 levels were indeed increased or decreased as intended after virus injection.
- 2) Experimental details were missing: what gender of the rats were used? What co-ordinations were used in the bilateral intra-hippocampal injection? How long after Sp1 injection when the behavioral tests were performed?
- 3) English needs attention.

Referee #2:

Guo et al. investigated the mechanism underlying vascular dementia focusing on the roles of Ang-II and TGF-b1 using animal and cellular models. The results suggested that Ang-II hypertension increased TGF-b1 and then upregulated phosphorylation of smad2/3 with nuclear translocation of smad complex, triggering the transcription of Sp1, which consequently damaged synaptic function.

I believe the observations are intriguing and may uncover the pathogenesis of Ang-II-related cognitive impairment and provide potential therapeutic targets for vascular dementia; however, the manuscript requires a revision for consideration for acceptance.

Comments:

1. The authors should confirm that the alterations in the TGF-b1 pathway in the brain was induced by direct action of Ang-II to neuronal cells or indirectly by other mechanisms like hypoperfusion and related decrease in energy / oxygen supply. They should measure the amount of Ang-II in the brain or CSF in the animal model.
2. Peripheral administration of Ang-II might induce activation of TGF-b1 pathway not only in the brain, but also in the peripheral organs. The authors should measure the amounts of TGF-b1 and related molecules in the plasma to examine the potential effects of TGF-b1 pathway activation in the peripheral organs on the neuronal functions.

Huazhong University
of Science and Technology
Tongji Medical College

Department of Pathophysiology

Xiaochuan Wang, Ph.D.

Professor

13 Hangkong Road

Wuhan, Hubei 430030

Telephone: 086-27-83692625 / Fax: 086-27-83693883

E-mail: wxch@mails.tjmu.edu.cn

Jan 23, 2025

Re: Manuscript EMBOR-2024-60245-T

“Elevated TGF- β 1 impaires cognitive function by activating smad2/3-Sp1 in AngII-related hypertension”

Dear Editor and Reviewers:

Kindly find online our revised manuscript EMBOR-2024-60245-T in which we have followed the suggestions and addressed the comments of the two referees. We are grateful to you for kindly giving us the opportunity to revise our manuscript. We also greatly appreciated the constructive critique and helpful suggestions of the referees. All revisions are highlighted with yellow color. The summary of our revisions and the point-by-point answers to the criticisms are as follows:

EDITOR COMMENTS

Thank you for the submission of your manuscript entitled 'Elevated TGF- β 1 impaires cognitive function by activating smad2/3-Sp1 in AngII-related hypertension' to EMBO Reports. I would like to apologize for the terribly slow process in this case. I admit that we struggle to find a suitable panel of experts referees in this case. Thank you for your notable patience in awaiting this decision.

I am glad to say that the referees are in principle favourably inclined on your work, but you will see that the current dataset is judged to be premature and we agree with the need for the requested further developments.

Briefly, we appreciate that you provide evidence for an angiotensin II > TGFbeta > Smad/SP1 pathway that leads to hypertension and vascular dementia in rat. Angiotensin II treatment increases

blood pressure, reduces hippocampal blood flow, leads to synaptic dysfunction and cognitive deficits, and is associated with elevated TGF- β 1. TGF- β 1 overexpression exacerbates cognitive impairment, while loss of function partially rescues the Ang II linked pathology. You show the effect is dependent on the canonical TGF- β 1 pathway i.e. Smad2/3, as well as SP1.

Both referees acknowledge that the findings are potentially interesting. Ref 1 notes 'the study was rigorously designed and well executed with solid data that supports the conclusion', and ref 2 'the observations are intriguing and may uncover the pathogenesis of Ang-II-related cognitive impairment'. However, ref 2 requires:

1) measure the amount of Ang-II in the brain and CSF.

Response: We really appreciate this point. Following this suggestion, we have measured the amount of Ang-II in the hippocampal tissues. The result was shown in Expanded View Figure 3A. Please see the revised manuscript!

Expanded View Figure 3

2) definitive additional experimental evidence to test whether AngII regulates TGF- β 1 directly or indirectly (and, if that is the case, how; for example by hypoperfusion and decreased energy / oxygen supply).

Response: These constructive comments are greatly appreciated. Following this suggestion, we have performed the primary hippocampal neuronal culture with or without AngII, oxygen-glucose deprivation (OGD) and CoCl₂ (a drug for chemical hypoxia model^{1,2}). We found that AngII does not affect the levels of TGF- β 1 in the primary hippocampal neurons (Fig.EV3D, E), while OGD and CoCl₂ both increased the levels of TGF- β 1 (Fig.EV3F-I), partially suggesting that AngII indirectly regulates TGF- β 1.

Please see the revised manuscript!

Expanded View Figure 3

Reference

1. Zhang, M., Ma, R. & Li, Q. Inhibitory Action of CoCl₂-Induced MCF-7 Cell Hypoxia Model of Breast Cancer And Its Influence On Vascular Endothelial Growth Factor. *J Biol Regul Homeost Agents* 29, 671-676 (2015).
2. Pecoraro, M., Pinto, A. & Popolo, A. Inhibition of Connexin 43 translocation on mitochondria accelerates CoCl₂-induced apoptotic response in a chemical model of hypoxia. *Toxicol In Vitro* 47, 120-128 (2018).
- 3) measure peripheral (plasma) Ang-II and TGF-β1 and demonstrate or exclude effects of TGF-β1 pathway activation in the peripheral organs on the described neuronal effects.

Response: We greatly appreciate this suggestion. Following this suggestion, we have detected peripheral (plasma) Ang-II and TGF-β1. The result was shown in Expanded View Figure 3B, C, supporting an activation of TGF-β1 pathway in the peripheral organs. However, the present data cannot absolutely make sure the effect of TGF-β1 pathway activation in the peripheral organs on the neuronal functions, which should be further investigated by downregulation of TGF-β1 in the peripheral organs in the future work Please see the revised manuscript!

Expanded View Figure 3

4) ref 1 requires data to directly show TGF-β1 expression in the CNS.

Response: Following this suggestion, we have measured TGF-β1 expression in the hippocampus (Figure 3E, F). Please see the revised manuscript !

Figure 3E, F

5) While not raised directly by the referees, I strongly encourage you to explore the molecular mechanistic connection between Smad2/3 and SP1 in the context of the transcriptional regulation as you suggest. Do these factors bind to relevant gene promoters together (we recommend ChIP or EMSAs if any relevant target genes are known). It is unclear what you mean with the statement '...promotes the entry of the phosphorylated smad2/3 complex into the nucleus, where they bind to Sp1 instead of the other transcription factors IRF-7, c-Jun, c-Fos.'. Please elaborate by adding experimental evidence that Smad2/3 phosphorylation regulates nuclear entry as claimed.

Response: We totally agree with this suggestion. Following this suggestion, we have further investigated the effect of Smad2/3 on the transcriptional regulation of Sp1.

Two plasmids with point mutations at the Smad3 Ser423/425 site, S423/425D (phosphorylation-activated state) and S423/425A (phosphorylation-inhibited state), were transfected into the N2A cell line. Then, we carried out ChIP-qPCR and detected the relationship between Sp1 and its relevant target genes *BACE1* and *HSP70*³⁻⁷, and found that mRNA levels of *HSP70* and

BACE1 significantly increased in the S423/425D group, while mRNA levels of *HSP70* were significantly decreased in the S423/425A group (Fig. 5N). Please see the revised manuscript!

Figure 5N

Reference

3. Nakano, M., Tsuchida, T., Mitsuishi, Y. & Nishimura, M. Nicotinic acetylcholine receptor activation induces BACE1 transcription via the phosphorylation and stabilization of nuclear SP1. *Neurosci Res* 203, 28-41 (2024).
4. Nong, W., Bao, C., Chen, Y. & Wei, Z. miR-212-3p attenuates neuroinflammation of rats with Alzheimer's disease via regulating the SP1/BACE1/NLRP3/Caspase-1 signaling pathway. *Bosn J Basic Med Sci* 22, 540-552 (2022).
5. Christensen, M.A. et al. Transcriptional regulation of BACE1, the beta-amyloid precursor protein beta-secretase, by Sp1. *Mol Cell Biol* 24, 865-874 (2004).
6. Bevilacqua, A., Fiorenza, M.T. & Mangia, F. Developmental activation of an episomic hsp70 gene promoter in two-cell mouse embryos by transcription factor Sp1. *Nucleic Acids Res* 25, 1333-1338 (1997).
7. Morgan, W.D. Transcription factor Sp1 binds to and activates a human hsp70 gene promoter. *Mol Cell Biol* 9, 4099-4104 (1989).

6) the referees also requests more details to describe both the behavioral and intra-hippocampal injection experiments. Please ensure the methods section is sufficiently detailed as as to render the experiments easily reproducible (we also encourage the link to protocols on e.g. protocols.io)

Response: Following this suggestion, we have provided more detail of the behavioral and intra-hippocampal injection experiments in the revised manuscript.

7) Please carefully revise the text with a view on optimizing the English and correcting typos e.g. in the abstract 'hypefusion'; in the conclusion, it is an overstatement (based on the current dataset) to claim '....triggering the transcription of Sp1 and ultimately leading to synaptic damage and cognitive impairments' > please provide support for this or revise the text.

Please simplify language where possible e.g. SP1 is a well know 'transcription factor' - considerably shorter than 'Specificity protein 1 (Sp1) is a widely DNA-binding protein expressed that both activates and inhibits gene transcription').

Response: We greatly appreciate this point. Following these suggestions, we have corrected language errors and typographical errors in this revised manuscript. We also invited Mr. Yacoubou Abdoul Razak Mahaman to do our paper proofreading in language and typographical design.

Referee #1:

Guo and colleagues established an animal model where AngII-induced hypertension caused synaptic and cognitive deficits in rats. They found TGFbeta1 expression phenocopying AngII treatment in terms of synaptic and cognitive deficits and inhibition of TGFbeta1 in AngII-treated rats rescued cognitive deficits. They further demonstrated that this was likely mediated by Smad2/3-Sp1 signaling. They concluded that TGFbeta1/Smad2/3/Sp1 axis plays a critical role in mediating AngII-induced cognitive deficits. Overall, the study was rigorously designed and well executed with solid data that supports the conclusion. There were several minor concerns:

1) While the authors demonstrated GFP expression in the hippocampus after virus injection, it was not confirmed whether TGF1beta1 levels were indeed increased or decreased as intended after virus injection.

Response: Following this suggestion, we have detected TGF- β 1 expression in the CNS in Figure 3E, F. Please see the revised manuscript !

Figure 3

2) Experimental details were missing: what gender of the rats were used? What co-ordinations were used in the bilateral intra-hippocampal injection? How long after Sp1 injection when the behavioral tests were performed?

Response: Our apologies! We have added the Experimental details in Material and Methods. Please see the revised manuscript!

3) English needs attention.

Response: We greatly appreciate this point. Following these suggestions, we have corrected language errors and typographical errors in this revised manuscript. We also invited Mr. Yacoubou Abdoul Razak Mahaman to do our paper proofreading in language and typographical design.

Referee #2:

Guo et al. investigated the mechanism underlying vascular dementia focusing on the roles of Ang-II and TGF-b1 using animal and cellular models. The results suggested that Ang-II hypertension increased TGF-b1 and then upregulated phosphorylation of smad2/3 with nuclear translocation of smad complex, triggering the transcription of Sp1, which consequently damaged synaptic function.

I believe the observations are intriguing and may uncover the pathogenesis of Ang-II-related cognitive impairment and provide potential therapeutic targets for vascular dementia; however, the manuscript requires a revision for consideration for acceptance.

Comments:

1. The authors should confirm that the alterations in the TGF-b1 pathway in the brain was induced by direct action of Ang-II to neuronal cells or indirectly by other mechanisms like hypoperfusion and

related decrease in energy / oxygen supply. They should measure the amount of Ang-II in the brain or CSF in the animal model.

Response: Following this suggestion, we have performed the primary hippocampal neuronal culture with or without AngII, oxygen-glucose deprivation (OGD) and CoCl₂ (a drug for chemical hypoxia model^{1, 2}). We found that AngII does not affect the levels of TGF-β1 in the primary hippocampal neurons (Fig.EV3D, E), while OGD and CoCl₂ both increased the levels of TGF-β1 (Fig.EV3F-I), partially suggesting that AngII indirectly regulates TGF-β1. We also have measured the amount of Ang-II in the hippocampal tissues. The result was shown in Fig.EV3A. Please see the revised manuscript!

Expanded View Figure 3

Reference

1. Zhang, M., Ma, R. & Li, Q. Inhibitory Action of Coc12-Induced Mcf-7 Cell Hypoxia Model of Breast Cancer And Its Influence On Vascular Endothelial Growth Factor. *J Biol Regul Homeost Agents* 29, 671-676 (2015).
2. Pecoraro, M., Pinto, A. & Popolo, A. Inhibition of Connexin 43 translocation on mitochondria accelerates CoCl₂-induced apoptotic response in a chemical model of hypoxia. *Toxicol In Vitro* 47, 120-128 (2018).2.

2. Peripheral administration of Ang-II might induce activation of TGF-b1 pathway not only in the brain, but also in the peripheral organs. The authors should measure the amounts of TGF-b1 and related molecules in the plasma to examine the potential effects of TGF-b1 pathway activation in the peripheral organs on the neuronal functions.

Response: We greatly appreciate this suggestion. Following this suggestion, we have detected peripheral (plasma) Ang-II and TGF- β 1 and the results showed that Ang-II treatment induced an increase in TGF- β 1 level (Fig EV3B, C), supporting an activation of TGF- β 1 pathway. However, the present data cannot absolutely make sure the effect of TGF- β 1 pathway activation in the peripheral organs on the neuronal functions, which should be further investigated by downregulation of TGF- β 1 in the peripheral organs in the future work. Please see the revised manuscript!

Expanded View Figure 3

We hope that you will find this revised manuscript acceptable for publication in *Embo reports*.

Sincerely,

Xiaochuan Wang, Ph.D.

Dear Dr. Wang

Thank you for the submission of your revised manuscript to our offices. We have now received the enclosed reports from the referees and I am please to say that they both recommended publication unconditionally.

At this stage, we only need to request a number of minor issues, as outlined below, for formal acceptance and expedited publication. In particular, we require that datasets are deposited and fully accessible in a stable, citable community database as a condition of publication.

Major issues:

- Please remove the sentence "The datasets used and/or analyzed during the present study are available from the corresponding author upon reasonable request." and include the specific URL for the GEO deposition in this section
- Ethics Approval and Consent to Participate (IRB approval with record number) should be part of the relevant Methods section
- Please provide a Reagents and Tools Table listing key reagents, experimental models, software and relevant equipment and including their sources and relevant identifiers. Please download and fill our Reagents and Tools Table template (.docx), which you can find in our author guidelines: <https://www.embopress.org/page/journal/14693178/authorguide#structuredmethods>. When submitting your revised manuscript, please do not include the Reagents and Tools Table in the Methods section of the manuscript but upload it as a separate file choosing the file type "Reagent Table". An example of a Method paper with Structured Methods can be found here: <https://www.embopress.org/doi/10.15252/msb.20178071>.

Style issues:

- Rename 'Availability of Data and Materials' to 'Data Availability' and place it after Acknowledgments;
- Rename 'Conflict of Interest' to 'Disclosure and Competing Interests Statement'
- Rename 'Materials and Methods' to 'Methods'
- Abbreviations section needs to be removed from the manuscript. Abbreviations should be defined in brackets after their first mention in the text, not in a list of abbreviations.
- Consent for Publication section should be removed from the ms
- Remove Author Contributions section from your manuscript file as you already provided the roles and contributions for each author upon resubmitting
- References: for references with more than 10 authors, et al should be used after the 10th name
- Remove 'the 'Funding' section heading and include the funding information under Acknowledgements. Please include funding details (e.g. numbers) as required, as this cannot be added later.
- Synopsis text: in addition to the 3 highlighting points, we also need a short, two-sentence summary of the manuscript (not more than 35 words).
- Synopsis image needs to be uploaded as a separate file in jpeg, TIFF or png format and needs to be exactly 550 pixels wide x 200-600 pixels high

DATA ISSUES:

- Please note that the exact p values should be provided in the legends of figures 1G-K; 2B, G, K, L, M; 2E, F, L, M, N, O, P, R; 4C-G, I, K, L, N, O, P; 5B, D, F, H, I, M, K, L, O; 7B, C, E, F, G, H, I, K, M, N; EV1 B, D; EV3 A, B, C, G, I; EV4 B.
- Please indicate the statistical test used for data analysis in the legends of figures 1D-K; 2B, D, G, I, K, L, M; 2B, E-R; 4C, D, E, F, G, I, K, L, N, O, P; 5B, D, F, H, I, M, K, N; 6C, E-O; 6E, F, G, J, K, L, M, O; 7B, C, E, F, G, H, I, J, K, M, N; EV1 B, D; EV2 B; EV3 A, B, C, E, G, I; EV4 B.
- Please note that information related to n (i.e. the number of independent replicates) is missing in the legends of figures 1B, C, D, F, G, H; 4B, EV1 B; EV2 B, EV4 B.

I look forward to a final revised version of your manuscript as soon as possible.

Best wishes,

Bernd Pulverer

~~~~~  
Bernd Pulverer, Ph.D.  
Chief Editor, EMBO Reports  
EMBO  
Meyerhofstrasse 1, D-69117 Heidelberg  
Tel: +4962218891501  
[bernd.pulverer@embo.org](mailto:bernd.pulverer@embo.org)  
~~~~~

Referee #1:

No more concerns.

Referee #2:

The authors addressed all the comments I suggested. No additional comments from my side.

Huazhong University
of Science and Technology
Tongji Medical College

Department of Pathophysiology

Xiaochuan Wang, Ph.D.

Professor

13 Hangkong Road

Wuhan, Hubei 430030

Telephone: 086-27-83692625 / Fax: 086-27-83693883

E-mail: wxch@mails.tjmu.edu.cn

Apr.7, 2025

Re: Manuscript EMBOR-2024-60245V2

Dear Editor:

Kindly find online our revised manuscript EMBOR-2024-60245V2 in which we have followed the suggestions and addressed the comments. We are grateful to you for kindly giving us the opportunity to revise our manuscript again. All revisions are highlighted with yellow color. The summary of our revisions and the point-by-point answers to the criticisms are as follows:

Major issues:

- Please remove the sentence "The datasets used and/or analyzed during the present study are available from the corresponding author upon reasonable request." and include the specific URL for the GEO deposition in this section

Response: Following this suggestion, we have removed the sentence "The datasets used and/or analyzed during the present study are available from the corresponding author upon reasonable request." and included the specific URL for the GEO deposition in this section.

This study includes the URL for the GEO deposition:

<https://www.ncbi.nlm.nih.gov/geo/query/acc.cgi?acc=GSE47529;>

<https://www.ncbi.nlm.nih.gov/geo/query/acc.cgi?acc=GSE126500;>

<https://www.ncbi.nlm.nih.gov/geo/query/acc.cgi?acc=GSE165771;>

<https://www.ncbi.nlm.nih.gov/geo/query/acc.cgi?acc=GSE37935;>

<https://www.ncbi.nlm.nih.gov/geo/query/acc.cgi?acc=GSE31628.>

- Ethics Approval and Consent to Participate (IRB approval with record number) should be part of the relevant Methods section

Response: We have added the IACUC number 4521 in the Methods section. Please see the

revised manuscript.

All animal experiments received approval from the Institutional Animal Care and Use Committee (IACUC) of Huazhong University of Science and Technology (IACUC NO.4521).

- Please provide a Reagents and Tools Table listing key reagents, experimental models, software and relevant equipment and including their sources and relevant identifiers. Please download and fill our Reagents and Tools Table template (.docx), which you can find in our author guidelines:

Response: Following this suggestion, we have provided a Reagents and Tools Table listing key reagents, experimental models, software and relevant equipment and including their sources and relevant identifiers.

Style issues:

- Rename 'Availability of Data and Materials' to 'Data Availability' and place it after Acknowledgments;

Response: Following this suggestion, we have renamed 'Availability of Data and Materials' to 'Data Availability' and placed it after Acknowledgments. Please see the revised manuscript.

- Rename 'Conflict of Interest' to 'Disclosure and Competing Interests Statement'

Response: Following this suggestion, we have renamed 'Conflict of Interest' to 'Disclosure and Competing Interests Statement'. Please see the revised manuscript.

- Rename 'Materials and Methods' to 'Methods'

Response: Following this suggestion, we have renamed 'Materials and Methods' to 'Methods'. Please see the revised manuscript.

- Abbreviations section needs to be removed from the manuscript. Abbreviations should be defined in brackets after their first mention in the text, not in a list of abbreviations.

Response: Following this suggestion, we have removed the ' Abbreviations ' section.

- Consent for Publication section should be removed from the ms

Response: Following this suggestion, we have removed the Consent for Publication section.

- Remove Author Contributions section from your manuscript file as you already provided the roles and contributions for each author upon resubmitting

Response: Following this suggestion, we have removed the Author Contributions section.

- References: for references with more than 10 authors, et al should be used after the 10th name

Response: Following this suggestion, we have revised the format of all references in accordance with the requirements.

- Remove 'the 'Funding' section heading and include the funding information under Acknowledgements. Please include funding details (e.g. numbers) as required, as this cannot be added later.

Response: Following this suggestion, we have removed 'the 'Funding' section heading and included the funding details under Acknowledgements.

- Synopsis text: in addition to the 3 highlighting points, we also need a short, two-sentence summary of the manuscript (not more than 35 words).

- Synopsis image needs to be uploaded as a separate file in jpeg, TIFF or png format and needs to be exactly 550 pixels wide x 200-600 pixels high

Highlights

1. Elevated TGF- β 1 is associated with synaptic and cognitive dysfunctions in AngII-related hypertension.

2. The nuclear entry of the TGF- β 1 downstream complex Smad2/3 is enhanced, and it binds to the transcription factor SP1 in the hippocampus of the AngII rats.

3. Downregulation of Sp1 improved TGF- β 1-induced synaptic and cognitive impairments.

Short Summary

Increased TGF- β 1 promotes the phosphorylation and nuclear translocation of Smad2/3, sequentially activating the transcription of Sp1, which is regarded as a key feature of AngII-induced synaptic and cognitive impairments.

C) Synopsis image

DATA ISSUES:

- Please note that the exact p values should be provided in the legends of figures 1G-K; 2B, G, K, L, M; 2E, F, L, M, N, O, P, R; 4C-G, I, K, L, N, O, P; 5B, D, F, H, I, M, K, L, O; 7B, C, E, F, G, H, I, K, M, N; EV1 B, D; EV3 A, B, C, G, I; EV4 B.

Response: Following this suggestion, we have provided the exact p values in the legends of these above figures.

- Please indicate the statistical test used for data analysis in the legends of figures 1D-K; 2B,

D, G, I, K, L, M; 2B, E-R; 4C, D, E, F, G, I, K, L, N, O, P; 5B, D, F, H, I, M, K, N; 6C, E-O; 6E, F, G, J, K, L, M, O; 7B, C, E, F, G, H, I, J, K, M, N; EV1 B, D; EV2 B; EV3 A, B, C, E, G, I; EV4 B.

Response: Following this suggestion, we have indicated the statistical test used for data analysis in the legends of the figures.

- Please note that information related to n (i.e. the number of independent replicates) is missing in the legends of figures 1B, C, D, F, G, H; 4B, EV1 B; EV2 B, EV4 B

Response: Our apologies! We have added the number of independent replicates in the legends of figures.

We hope that you will find this revised manuscript acceptable for publication in *EMBO Reports*.

Sincerely,

Xiaochuan Wang, Ph.D.

Prof. Xiaochuan Wang
Tongji Medical College, HUST
Pathophysiology
13 Hangkong Road
Wuhan, Hubei 430030
China

Dear Prof. Wang,

I am pleased to inform you that your manuscript has been accepted for publication in EMBO reports. Your manuscript will be processed for publication by EMBO Press. It will be copy edited and you will receive page proofs prior to publication. Please note that you will be contacted by Springer Nature Author Services to complete licensing and payment information.

Yours sincerely,

Bernd Pulverer

~~~~~  
Bernd Pulverer, Ph.D.  
Chief Editor, EMBO Reports  
EMBO  
Meyerohofstrasse 1, D-69117 Heidelberg  
Tel: +4962218891501  
[bernd.pulverer@embo.org](mailto:bernd.pulverer@embo.org)  
~~~~~
